# White-light activatable organic NIR-II luminescence nanomaterials for imaging-guided surgery

Chunbin Li[1,5], Jian Du[2,5], Guoyu Jiang[1,5], Jianye Gong[1], Yue Zhang[1], Mengfan Yao[1], Jianguo Wang[1] ✉, Limin Wu[1,3] ✉ & Ben Zhong Tang[4]

While second near-infrared (NIR-II) fluorescence imaging is a promising tool for real-time surveillance of surgical operations, the previously reported organic NIR-II luminescent materials for in vivo imaging are predominantly activated by expensive lasers or X-ray with high power and poor illumination homogeneity, which significantly limits their clinical applications. Here we report a white-light activatable NIR-II organic imaging agent by taking advantages of the strong intramolecular/intermolecular D-A interactions of conjugated Y6CT molecules in nanoparticles (Y6CT-NPs), with the brightness of as high as 13315.1, which is over two times that of the brightest laser-activated NIR-II organic contrast agents reported thus far. Upon white-light activation, Y6CT-NPs can achieve not only in vivo imaging of hepatic ischemia reperfusion, but also real-time monitoring of kidney transplantation surgery. During the surgery, identification of the renal vasculature, post-reconstruction assessment of renal allograft vascular integrity, and blood supply analysis of the ureter can be vividly depicted by using Y6CT-NPs with high signal-to-noise ratios upon clinical laparoscopic LED white-light activation. Our work provides efficient molecular design guidelines towards white-light activatable imaging agent and highlights an opportunity for precision imaging theranostics.

As an integral and indispensable part of healthcare, surgery focuses on curing or diagnosing disease in patients[1], such as the removal of diseased tissues, repair of damages, transplantation of organs, etc. A crucial issue for surgeons in clinical procedures is to distinguish damaged or diseased tissues from healthy ones to avoid wrong operations and to preserve normal functions of target organs as much as possible, which leads to an urgent need of real-time monitoring approaches. The second near-infrared (NIR-II, 1000–1700 nm) imaging with high temporal-spatial resolution, deep penetration, and low autofluorescence interference, has gained extensive interest in the fields of biological and medical sciences[2–5]. Intraoperatively, NIR-II imaging can provide useful information to surgeons including labeling cancer localization[6–9], tracking lymph nodes[10], assessing surgical margins[11–13], and mapping key anatomical structures[14–17]. However, the previously reported organic NIR-II imaging contrast agents are mainly activated by expensive high-power lasers or X-ray[18–24]. Limited absorption of laser light with single wavelength will inevitably discount the output signals and the high power of laser or X-ray may bring about potential biological damage in clinical practice[25,26]. Worse still, narrow bandwidth lasers cannot provide homogeneous illumination and

[1]College of Chemistry and Chemical Engineering, College of Energy Material and Chemistry, Inner Mongolia Key Laboratory of Fine Organic Synthesis, Inner Mongolia University, Hohhot 010021, China. [2]Department of Urology, The First Affiliated Hospital of Shandong First Medical University, Jinan 250000 Shandong, China. [3]Department of Materials Science and State Key Laboratory of Molecular Engineering of Polymers, Fudan University, Shanghai 200433, China. [4]School of Science and Engineering, Shenzhen Institute of Aggregate Science and Technology, The Chinese University of Hong Kong, Shenzhen (CUHK-Shenzhen), Shenzhen 518172 Guangdong, China. [5]These authors contributed equally: Chunbin Li, Jian Du, Guoyu Jiang. ✉ e-mail: wangjg@iccas.ac.cn; wlm@imu.edu.cn

therefore increase the operating difficulties for surgeons. By contrast, as a secure broad continuous spectra, white-light, (especially clinically used laparoscopic-light and surgical light in the operating room) would be a powerful and ideal alternative to address these issues[27]. Unfortunately, NIR-II imaging contrast agent that can be activated by natural white-light has never been reported so far due to the lack of efficient design strategy.

To ensure NIR-II fluorophores can be effectively excited by white-light, they should possess a continuous and high molar extinction coefficient ($\varepsilon$) in the wavelength range of 400–780 nm, as well as a high quantum yield (QY). Organic small conjugated molecules can usually span the visible, NIR-I and even NIR-II region emission windows by precise regulation of their molecular structures[28–31], such as the extension of molecular conjugation backbone and the improvement of donor-acceptor (D-A) interaction[32,33]. Although some small organic NIR-II fluorophores including D-π-A, D-A-D, and A-D-A type have been developed for imaging-guided surgery in the past few years[6,30,34–39], they were majorly activated upon laser or X-ray due to their low-absorptivity in the visible region. Y6CT, as an A-D-A type conjugated molecule with high $\varepsilon$ in the visible and NIR ranges, displays high promise in NIR-II imaging-guided surgery but has never been investigated.

In this work, by taking advantages of the strong intramolecular/intermolecular D-A interactions in aggregate state, we report a white-light activatable NIR-II organic imaging agent through nano-partialization of Y6CT molecules. The obtained Y6CT nanoparticles (Y6CT-NPs) display NIR-II emission brightness of as high as 13,315.1, which is even more than two times that of the brightest laser-activated organic NIR-II small molecules reported so far. Experimental results demonstrate Y6CT-NPs can be efficiently applied for white-light activatable in vivo NIR-II bioimaging of hepatic ischemia reperfusion and real-time monitoring of kidney transplantation surgery.

## Results

### Design and characterization of Y6CT

Firstly, a conjugated planar molecule with A-D-A structure, namely Y6CT, was synthesized with the largely conjugated dithienothiophen[3.2-b]-pyrrolobenzothiadiazole as an electron-donating unit, and 2-(4-oxocyclopenta[c]thiophen-6-ylidene) propanedinitrile as the electron-withdrawing unit. Synthetic details were summarized in Supplementary Fig. 1. All the results of [1]H NMR, [13]C NMR spectroscopy, and high-resolution mass spectrometry (HRMS) confirm the Y6CT molecule has been successfully synthesized (Supplementary Figs. 2–4).

To further enhance the visible absorption and optimize the biocompatibility and biostability of the hydrophobic molecule for biological imaging applications, Y6CT was formulated into nanoparticles by taking advantages of the strong intramolecular/intermolecular D-A interactions in aggregate state using 1,2-distearoyl-sn-glycero-3-phosphoethanolamine-N-[methoxy(polyethyleneglycol)] (DSPE-mPEG$_{2000}$) as the encapsulation matrix (Fig. 1a). High-resolution transmission electron microscopy images display Y6CT-NPs have uniform core-shell morphology with a spherical phase of ~176 nm determined by dynamic light scattering (Fig. 1b). Additionally, the stability of Y6CT-NPs in long-term storage was further analyzed. As shown in Supplementary Fig. 5, the diameter of Y6CT-NPs experienced negligible variations over a 14-day period in a PBS solution, indicating a favorable stability. From the UV-vis-NIR and photoluminescence (PL) spectroscopy, Y6CT exhibits strong absorption at 500–800 nm in CHCl$_3$ and THF solution. And notably, in nanoparticle state, Y6CT-NPs maintained excellent light-harvesting capability with broad absorption extended to 1000 nm (Fig. 1c). Compared with Y6CT solution state, the Y6CT-NPs show much broader and obviously redshifted absorption band centered at 798 nm and an $\varepsilon$ value of $8.26 \times 10^4$ L mol$^{-1}$ cm$^{-1}$.

The high light-harvesting ability of Y6CT and Y6CT-NPs in visible and NIR regions prompted us to explore the fluorescence behavior after white-light activation. A clinical laparoscopic LED light was used as the excitation light for the following fluorescence spectral analysis. Both Y6CT in solution and nanoparticle state exhibit high NIR fluorescence emission upon laparoscopic LED light irradiation (Fig. 1d). Particularly, Y6CT-NPs present a bright NIR-II emission extending to 1400 nm with two peaks of 947 nm and 1030 nm in aqueous solution, significantly redshifted compared to Y6CT solution state. In order to elucidate this phenomenon, absorption and emission spectra of Y6CT in mixed solution with varied THF/water ratios were recorded. As shown in Supplementary Fig. 6, Y6CT demonstrated a redshift in the absorption peak from 706 to 789 nm and a widening of the absorption band with increasing water fractions. Likewise, the emission peak was also redshifted from 790 to 1034 nm. Importantly, the spectra of Y6CT in the aggregated state closely resemble those of the nanoparticle state, highlighting that the redshift and broadening of Y6CT-NPs' spectra primarily depend on efficient molecular stacking. In addition, the fluorescence spectra of FY6-NPs, as a nanomaterial outlined in our previous study[39], was investigated under white-light activation. Although FY6-NPs demonstrate NIR-II emission upon white-light excitation, their PL intensity at maximum emission wavelength is only 0.4-times that of Y6CT-NPs (Supplementary Fig. 7). These results afford the Y6CT-NPs a probability for NIR-II imaging in biological systems under white-light activation.

To evaluate the luminescence efficiency of Y6CT-NPs, their QY was determined using commercially available NIR-II contrast agent 4-[2-[2-chloro-3-[2-(2-phenyl-2H-thiochromen-4-yl)ethenyl]cyclohex-2-en-1-ylidene]ethylidene]-2-phenylthiochromene (IR26) (QY = 0.5%) in 1,2-dichloroethane (DCE) as the reference[24,39]. As shown in Fig. 1e and Supplementary Fig. 8, Y6CT-NPs display an impressive QY of 16.12% in the 850–1400 nm range in aqueous solution (QY of 8.62% in the 1000–1400 nm range) using 808 nm laser as excitation source, which is ~32-times that of IR26 and ~4-times higher than our previously reported FY6-NPs (QY of 4.08% and 2.24% in the spectral ranges of 850–1400 nm and 1000–1400 nm, respectively)[39]. Furthermore, a comparison of the photophysical properties of Y6CT-NPs with those previously reported organic NIR-II small molecule materials was carried out and Y6CT-NPs evidently stand out of these materials (Fig. 1f and Supplementary Table 1). The NIR-II fluorescence brightness, an integral factor in the imaging performance of fluorescent molecules, was assessed by calculating the QY × $\varepsilon$ values. Notably, Y6CT-NPs reveal the brightness values of as high as 13,315.1 in the spectral ranges of 850–1400 nm, which is more than two time that of the brightest NIR-II organic small molecule nanomaterials reported thus far[40]. More importantly, Y6CT-NPs show excellent emission with a NIR-II brightness values (>1000 nm) of 7120.1, far exceeding those of previously reported NIR-II organic small molecule nanomaterials (Fig. 1f and Supplementary Table 1). The bright NIR-II emission of Y6CT-NPs can greatly improve signal-to-noise ratio (SNR) for high-resolution NIR-II imaging under white-light activation.

To further validate the advantages of Y6CT-NPs in biological application, their NIR-II imaging property was compared with that of FY6-NPs and several commercial laser-activated NIR-II dyes, including indocyanine green (ICG), IR26, and 1-butyl-2-[2-[3-[2-(1-butyl-6-chlorobenz[cd]indol-2(1H)-ylidene)ethylidene]-2-chloro-1-cyclohexen-1-yl]ethenyl]-6-chlorobenz[cd]indolium tetrafluoroborate (IR1048). From Fig. 1g and Supplementary Fig. 9, one can see that all these commercial dyes show undetectable fluorescence signals upon white-light illumination. Very impressively, Y6CT-NPs exhibit a bright NIR-II emission, with an intensity of over 25-times higher than that of commercial dyes and nearly 4-times stronger than that of FY6-NPs, suggesting the great potential for in vivo imaging under white-light activation. Given the high NIR-II brightness of Y6CT-NPs, we further estimated its photostability in aqueous solution. As shown in Fig. 1h, Y6CT-NPs

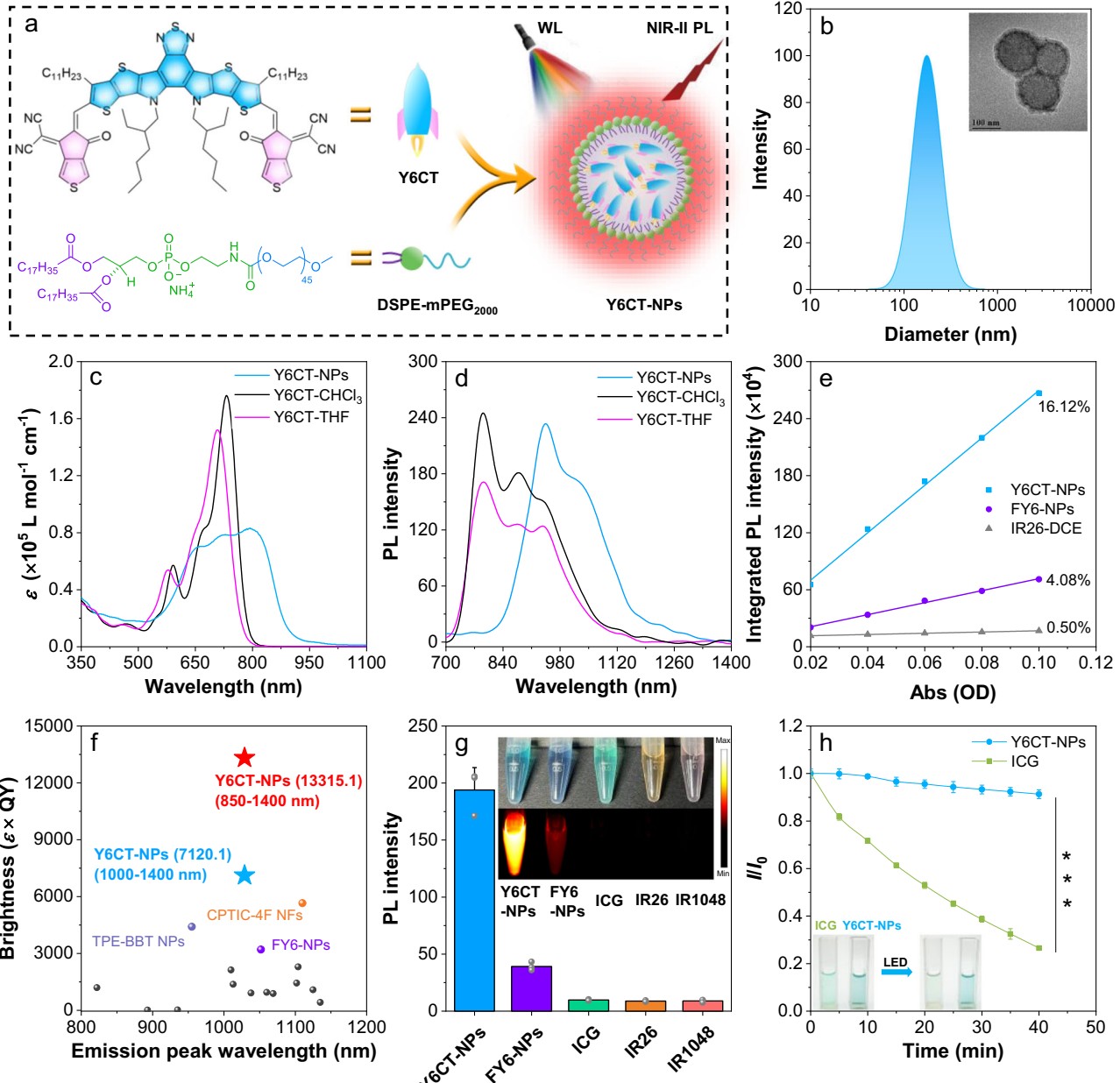

**Fig. 1 | Photophysical properties of Y6CT-NPs. a** Schematic illustration of the formulation of Y6CT-NPs. WL white-light. **b** Dynamic light scattering and TEM image of Y6CT-NPs. Repeated for three times in independent experiments. Molar extinction coefficient (**c**) and fluorescence spectra (**d**) of Y6CT in nanoparticle and solution state ($E_x$ = Laparoscopic-light, 643 nm long-pass filter). **e** The linear fitting of the integrated PL intensity vs. the absorbance values of Y6CT-NPs in deionized water, FY6-NPs in deionized water and IR26 in DCE ($E_x$ = 808 nm laser). **f** The NIR-II fluorescence brightness ($\varepsilon \times$ QY) of different NPs. **g** The mean fluorescence intensity of Y6CT-NPs in water, FY6-NPs in water, ICG in water, IR26 in DCE, and IR1048 in DCE with the same concentration (10 μM). Data were presented as mean ± SD derived from $n = 3$ independent samples per group. The inset shows their NIR-II fluorescent images under white-light illumination (16.5 mW cm$^{-2}$) and corresponding photographs under daylight. **h** Photostability of ICG and Y6CT-NPs (10 μM) in water upon continuous laparoscopic-light (50 mW cm$^{-2}$) illumination for 40 min. Data were presented as mean ± SD derived from $n = 3$ independent samples per group. $p$ value indicates the significant difference. $p = 0.0002$, ***$p < 0.001$, and statistical significance was assessed via an unpaired two-sided $t$-test method. $I$ represents the PL intensity of the samples at different time, while $I_0$ indicates the initial intensity at 0 min. The inset shows their corresponding photographs under daylight before and after laparoscopic-light activation.

---

demonstrate superior photostability of more than 90% retention of the initial brightness upon constant laparoscopic-light illumination for 30 min, whereas ICG experiences severe photodegradation, with the intensity dropping to less than 30% of the initial value. In addition, Y6CT-NPs maintained high brightness in PBS, fetal bovine serum (FBS: QY = 14.48%), and urine for 12 h at 37 °C without obvious variations, offering favorable stability for NIR-II imaging in biological systems (Supplementary Figs. 10 and 11). Furthermore, to exclude the unfavorable photothermal injury, the photothermal efficiency of Y6CT-NPs

was assessed under white-light activation. As depicted in Supplementary Fig. 12, the temperature variation of Y6CT-NPs remained consistently stable without notable increase under continuous exposure for 6 min, even at high concentrations of 100 μM, emphasizing favorable safety profile as a contrast agent for imaging purposes.

## Crystal analysis and theoretical calculations
The photophysical properties of a fluorophore are highly dependent on its single molecular structure and molecular packing mode. To

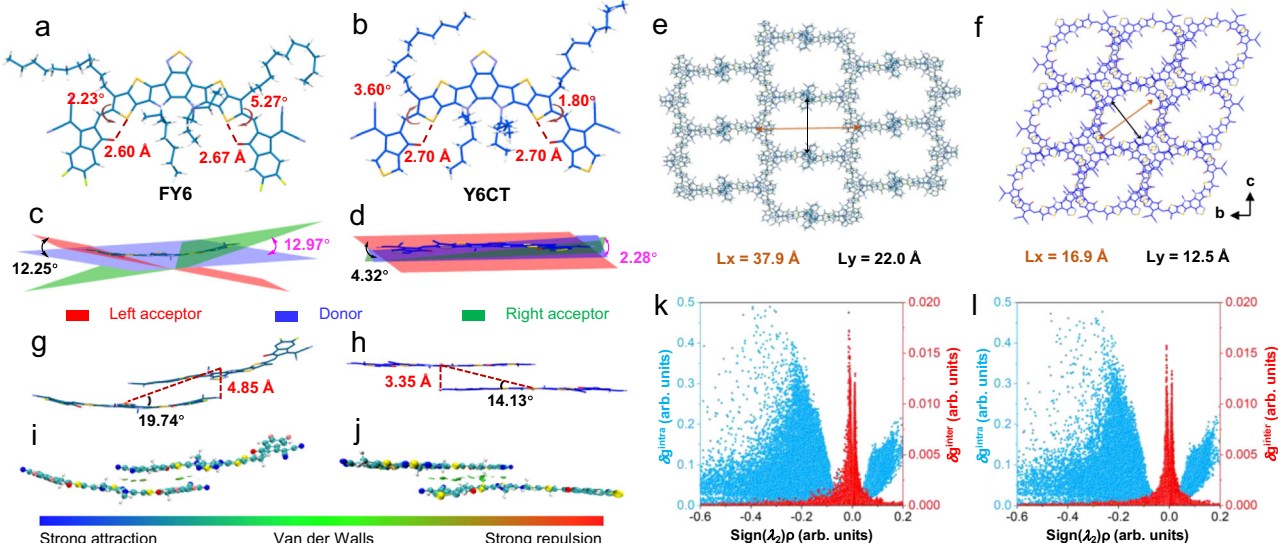

**Fig. 2 | Crystal analysis and theoretical calculations.** S···O distances and donor-acceptor torsions (**a**, **b**), donor-acceptor dihedrals (**c**, **d**), crystal packing (**e**, **f**), dimer structure (**g**, **h**) by single-crystal analysis of FY6 and Y6CT. The visualized isosurfaces of the IGM analysis (**i**, **j**) and 2D plot of $\delta g^{intra}$ (blue) and $\delta g^{inter}$ (red) for dimer structure (**k**, **l**) in FY6 and Y6CT.

investigate this, single crystals of Y6CT were obtained by slow vapor diffusion. The detailed metrical parameters of Y6CT (CCDC: 2302102) were presented in Supplementary Table 2. In order to disclose a deep understanding of the high fluorescence emission of Y6CT, previously reported FY6 molecule and its single-crystal data were used for comparison[39,41]. As shown in Fig. 2a, b, two fluorophores have essentially planar structures enforced by the fully fused core and the S···O conformational locking. In FY6 crystals, the torsion angles between end-groups and core are 2.23° and 5.27°, and their dihedral angles are 12.25° and 12.97°, respectively, with the corresponding S···O interaction distances of 2.60 and 2.67 Å (Fig. 2a, c). Notably, smaller torsion angles of 3.60° and 1.80° are observed in Y6CT, and its dihedral angles are decreased to 4.32° and 2.28°, with similar S···O interaction distances of 2.70 Å (Fig. 2b, d), which may suppress non-radiative transition processes from rotation or vibration. In addition, the crystal packing pattern can also profoundly impact the optical properties in aggregated states. Both FY6 and Y6CT display a grid-like packing mode, similar to most of the A-D-A structural molecules[42]. The strength of molecular interactions is closely related with the framework size. The smaller the size, the stronger the interactions are. FY6 shows an ellipsoidal network packing structure with Lx = 37.9 Å (major axis) and Ly = 22.0 Å (minor axis), while Y6CT exhibits a rectangular network structure with a smaller size of Lx = 16.9 Å and Ly = 12.5 Å, thus demonstrating stronger intermolecular interactions in comparison to FY6 (Fig. 2e, f). To gain a more comprehensive understanding, the effect of intermolecular interactions between two closely contacted molecules were further analyzed. For FY6, the interlamellar distance between the two stacking molecules is 4.85 Å with a slip angle of 19.74°, which is much lower than 54.7°, indicating a J-aggregation packing behavior (Fig. 2g). Compared with FY6, Y6CT shows a smaller adjacent intermolecular distance of 3.35 Å with a slip angle of 14.13°, demonstrating a tighter molecular packing (Fig. 2h). This arrangement promotes constructive coupling of excited-state transition dipoles, leading to an enhanced QY of Y6CT in the aggregated state. Besides, IGM analysis was also conducted with the Multiwfn program, and the VMD program was used to render the isosurfaces of weak interactions visible[43–46]. Figure 2i, j demonstrates that a green region is present between the two stacking molecules in both FY6 and Y6CT, indicating the formation of strong van der Waals interactions. Notably, Y6CT has a larger green region than FY6, corresponding to relatively stronger

intermolecular D-A interactions. This is further confirmed by the 2D plot of $\delta g^{inter}$ (the descriptor for defining intermolecular interaction regions) (Fig. 2k, l), which is in accordance with the results of intermolecular interaction distance analysis (Fig. 2g, h). Thus, it can be concluded that the tight J-aggregation packing and the strong intermolecular interactions together endow Y6CT with high QY and superior $\varepsilon$ in aggregate states, resulting in the high brightness in nanoparticles.

## In vivo angiography
Given the high brightness and good photostability of Y6CT-NPs upon white-light activation, in vivo fluorescence imaging in the NIR-II window was performed. First, the cytotoxicity of Y6CT-NPs was evaluated by the standard thiazolyl blue tetrazolium bromide (MTT) assays at various concentrations. As revealed in Supplementary Fig. 13, Y6CT-NPs had low cytotoxicity towards LO2 cells and NIH-3T3 cells in the dark, with cell viability after 24 h of incubation remaining above 95% even at high NPs concentration (100 µmol mL⁻¹), indicating that Y6CT-NPs are compatible with normal cells. When exposed to white-light, these cells also retained a high level of viability (≥ 90%), affirming negligible phototoxicity of Y6CT-NPs. In addition, systemic toxicity evaluation revealed that Y6CT-NPs had negligible organ and hematological toxicity in vivo (Supplementary Figs. 14 and 15 and Supplementary Tables 3 and 4). To visualize the imaging effect in vivo, the whole-body fluorescence angiography of Y6CT-NPs was evaluated in mice utilizing a low-power white-light illumination as excitation light. As shown in Fig. 3a, the vascular structures with many tiny capillary vessels branching from larger vessels in the abdomen is clearly delineated by the NIR-II fluorescence signal with different long-pass (LP) filters (900, 1000, and 1100 nm). In particular, the NIR-II images with 1100 nm LP filter exhibit the highest resolution, characterized by its larger SNR (1.79) and narrower full width at half maximum (FWHM, 173.8 µm), surpassing those of the images with 900 or 1000 nm LP filter (Fig. 3b and Supplementary Fig. 16). Furthermore, a comparison of in vivo imaging was performed between Y6CT-NPs and the FDA-approved commercial dye ICG (Supplementary Fig. 17). Unfortunately, the NIR-II fluorescence signal of ICG is almost undetectable under white-light excitation. Additionally, in vivo imaging using ICG under 808 nm laser irradiation was also performed for comparison. Although detectable signals were captured as displayed in Supplementary

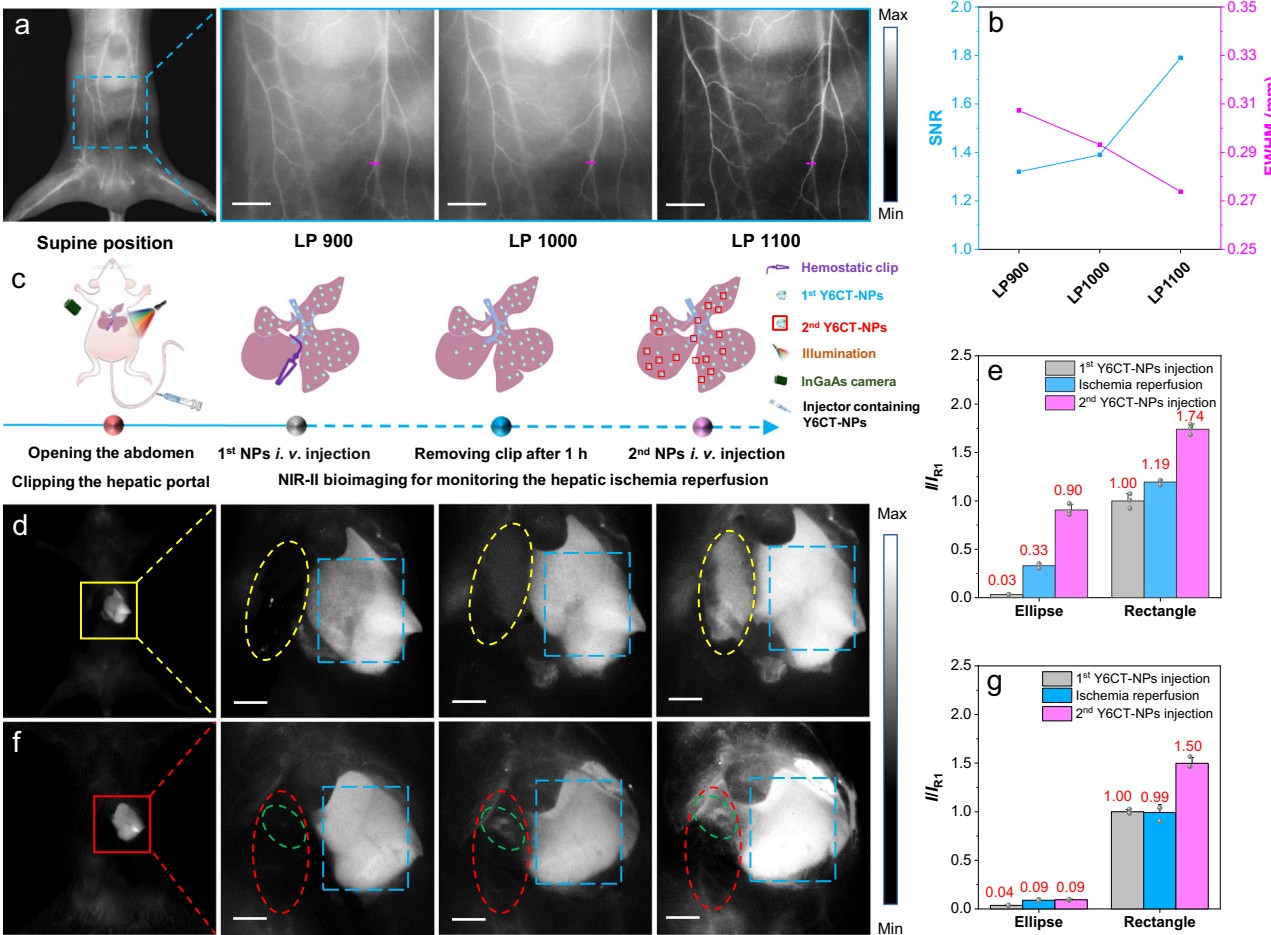

**Fig. 3 | White-light activated Y6CT-NPs for in vivo angiography and monitoring of hepatic ischemia reperfusion. a** NIR-II fluorescence imaging of blood vessels of BALB/c mice in supine positions after i.v. injection of 100 μL Y6CT-NPs (300 μM) under white-light illumination activation (16.5 mW cm$^{-2}$). Repeated for three times in independent experiments. **b** The cross-sectional signal-to-noise ratios and full width at half maximum was measured along the pink line. **c** Schematic diagram of hepatic ischemia reperfusion process. Fluorescence imaging of the liver with complete blood vessels (**d**) and corresponding fluorescence intensity ratio ($I/I_{R1}$) in different processes (**e**) of hepatic ischemia reperfusion after i.v. injection of Y6CT-NPs (300 μM, 1st injection 100 μL, 2nd injection 50 μL). Data were presented as mean ± SD derived from $n = 3$ independent samples per group. Fluorescence imaging of the liver with injured blood vessels (**f**) and corresponding fluorescence intensity ratio ($I/I_{R1}$) in different processes (**g**) of hepatic ischemia reperfusion after i.v. injection of Y6CT-NPs (300 μM, 1st injection 100 μL, 2nd injection 50 μL). The yellow and red ellipses denote the ischemic portion of liver, the blue rectangles denote the healthy liver, and the green ellipses denote the location of hepatic portal. Data were presented as mean ± SD derived from $n = 3$ independent samples per group. $I$ represent the average fluorescence intensity of the analyzed area at different stages, while $I_{R1}$ indicates the initial intensity of the rectangular region after 1st injection of Y6CT-NPs. $E_x$ = white-light illumination (16.5 mW cm$^{-2}$). Scale bar: 5 mm.

Fig. 17, ICG exhibited notably lower SNR (1.12) than that achieved with white-light activated Y6CT-NPs (1.79), further affirming the superiority of Y6CT-NPs as a white-light activatable contrast agent. For the purpose of confirming the blood half-life of Y6CT-NPs, a series of blood samples were collected over time, with the fluorescence intensity of each sample subsequently measured, as depicted in Supplementary Fig. 18. The results displayed that the blood half-life of Y6CT-NPs is 1.15 h. In addition, the short-term metabolic behavior of Y6CT in vivo was assessed by conducting ex vivo biodistribution studies in mice on the 1st, 3rd, 5th, and 7th days post-administration of Y6CT-NPs (Supplementary Fig. 19). The results revealed bright NIR-II fluorescence signals in the liver and spleen, indicating the principal metabolic pathways of Y6CT-NPs through these organs. A subsequent gradual decrease of fluorescence intensity in these two organs suggested the progressive metabolism of Y6CT-NPs via the liver and spleen.

**Monitoring hepatic ischemia reperfusion**

Hepatic ischemia reperfusion injury (HIRI) is an unavoidable process in liver transplantation, hepatic resection, and other surgeries, which can lead to early failure of the transplanted liver, tissue damage, organ rejection, and even liver failure[47]. Consequently, it is important to observe the process of hepatic ischemia reperfusion (HIR) and to promote blood reperfusion[48,49]. A frequent obstruction to this procedure is a blockade or injury to the hepatic portal. The NIR-II fluorescence imaging of Y6CT-NPs was conducted to monitor the HIR process in living mice utilizing a low-power white-light illumination as excitation light (Fig. 3c). Upon tail vein (*i.v.*) injection of Y6CT-NPs, a significant NIR-II fluorescence signal was observed in the healthy part (blue rectangle region) of the liver (the fluorescence intensity was denoted as $I_{R1}$), while the left inferior lobe (ischemic site, yellow ellipse) shows a negligible fluorescence signal (0.03 $I_{R1}$) due to the hepatic portal being blocked by hemostatic forceps (Fig. 3d, e). Removing the hemostatic forceps after 1-h post-injection, we can see that the NIR-II fluorescence of the yellow area gradually increases to 0.33 $I_{R1}$ within 10 min, indicating the revival of blood flow to the ischemic portion. However, its fluorescence intensity is much lower than that of the healthy liver (rectangular region), which is attributed to the metabolism of Y6CT-NPs. Following another injection of Y6CT-NPs, the

fluorescence intensity of the yellow area increases rapidly from 0.33 $I_{R1}$ to 0.90 $I_{R1}$, similar to the variation of the healthy liver (1.19 $I_{R1}$ vs. 1.74 $I_{R1}$), thus providing further evidence that the blood supply function has been restored. By contrast, in Fig. 3f, g, the left inferior lobe of the liver (red ellipse) displays relatively unchanged fluorescence intensity after the three-step processes (0.04 $I_{R1}$, 0.09 $I_{R1}$ and 0.09 $I_{R1}$), probably due to the damage of the hepatic portal, which prevents the reperfusion of blood. In addition, a marked increase of the fluorescence signal can be observed in the leakage segment of the blood vessels (green ellipse, Fig. 3f), thereby aiding doctors in determining the location of vascular injuries. These findings indicate that NIR-II fluorescence imaging of the liver with Y6CT-NPs could be employed not only to report HIR process during surgery but also to further verify the location of vascular injuries.

### Imaging-guided surgery of kidney transplant

Kidney transplantation is the optimal choice for patients with end-stage renal failure (ESRD) which has been demonstrated to improve survival rate and the wellbeing of patients in comparison to hemo-dialysis and peritoneal dialysis[50]. Intraoperative accurate identification of the renal vasculature and post-reconstruction assessment of renal allograft vascular integrity are both essential for the success of kidney transplantation. Before utilization, the negligible systemic toxicity of Y6CT-NPs was confirmed in rabbit via the histological investigation of organs and hematological analysis (Supplementary Fig. 20 and Sup-plementary Tables 5 and 6). To assess the performance of Y6CT-NPs in NIR-II fluorescence imaging of the renal vasculature, the in situ model of transplanted kidney was conducted in white rabbits to simulate real clinical scenarios (Fig. 4a). Y6CT-NPs was employed to analyze the positioning of the renal artery and vein of the donor kidney region. As shown in Fig. 4b and Supplementary Fig. 21, the renal artery (red arrow) and vein (blue arrow) are speedily visualized with high-resolution NIR-II fluorescence imaging (SNR = 3.26 ~ 3.39) after intravenous infusion of Y6CT-NPs under the laparoscopic-light activation. Notably, the boundaries of the renal arteries and veins remain clear and easily identifiable when the kidney is rotated 180° counterclockwise around the renal arteries and veins (Fig. 4b-iv). This suggests that Y6CT-NPs can visualize renal vasculature well in different orientations under laparoscopic-light source for further application in guiding donor nephrectomy (Supplementary Movie 1). For comparison, ICG was incapable of providing detectable signals after activated by white-light (Supplementary Fig. 22). Even under 808 nm laser irradiation, ICG exhibited poor performance compared to white-light activated Y6CT-NPs, thus once again confirming the good imaging capabilities of Y6CT-NPs as a white-light activated contrast agent for in vivo imaging.

Subsequently, the transplanted kidney model was established by connecting the arteriovenous vein of the donor kidney to that of the recipient using polyglactin sutures. After injection of Y6CT-NPs, a bright NIR-II fluorescence signal is detected first in the renal arteries at the instant of releasing the hemostatic forceps (Fig. 4c-ii, red arrow), and the outline of the kidney and renal vein then emerges and gra-dually becomes clear (Fig. 4c-iii, blue arrow). Excitedly, no aberrant fluorescence signal at the suture site of the vein and artery is observed, indicating a successful anastomosis. Besides, the bright NIR-II emission is uniformly distributed on the renal artery and vein even rotating the kidney 180° counterclockwise (Fig. 4c-iv), which is consistent with angiography of donor renal before the transplantation. The data indicate the feasibility of Y6CT-NPs assisting NIR-II renal angiography in checking the integrity of vascular anastomosis, allowing surgeons to make timely response (Supplementary Movie 2). However, some complications often accompany renal vascular anastomoses in kidney transplantation, which is difficult for surgeons to distinguish with the bright-field image alone. In order to assess the potential of Y6CT-NPs to promptly diagnose abnormal anastomosis of graft vasculatures, three different models of vascular stenosis including artery stenosis,

venous stenosis, and venous torsion were constructed. For the artery stenosis model, as displayed in Fig. 4d-ii, only the distal renal artery (red arrows) is light up by the NIR-II fluorescence after intravenous injection of Y6CT-NPs, while the transplanted renal artery is unobser-vable (green arrow) due to the stenosis of the transplanted renal artery anastomosis, which prevents smooth passage of blood flow. An effort to open the flow of the renal artery anastomosis by manipulating the graft renal vasculature was made, however, only few blood passes through the proximal graft renal artery with a weak fluorescence signal (Fig. 4d-iii, green arrow), indicating the function loss of graft artery. Furthermore, the graft renal vein becomes light-up because of the inferior vena cava regurgitation (Fig. 4d-iii, iv, blue arrow). For the venous stenosis, the fluorescence is slowly growing in the renal artery and kidney first (Fig. 4e-ii, green arrow and red arrow), and then blocked at the graft renal vein vascularization anastomosis (Fig. 4e-ii, blue arrow, pink arrow). Notably, the NIR-II fluorescence of the grafted kidney is slightly increased after adjusting the graft renal vasculature, but much lower than that of smooth vasculature (Fig. 4e-iii, iv), man-ifesting the graft renal vein anastomosis has been partially occluded. For venous torsion, only few fluorescence signals are found in renal vasculature due to the distortion of the graft renal vein anastomosis, which hinders the flow of blood with Y6CT-NPs back through the graft renal vein vascularization anastomosis into the inferior vena cava (Fig. 4f-iii, blue arrow, pink arrow). Importantly, in the case of venous torsion, blood flow can be easily restored to normal state by reor-ienting the graft renal vasculature (Fig. 4f-iv). In short, abnormal ana-stomosis of the grafted renal vasculature could be easily diagnosed and repaired by Y6CT-NPs based angiography under the laparoscopic-light activation.

Transplant kidney ureters are particularly vulnerable to poor blood supply, which easily causes non-healing of the anastomosis between the ureter and the bladder, urine retention to the pelvis, and even death due to infection in severe cases. To demonstrate the potential of Y6CY-NPs, two models of transplanted kidney ureters with good and poor blood supply were constructed. Following intravenous injection of Y6CT-NPs, the left ureter and the proximal end of the right ureter were immediately light up, while the distal end of the right ureter was invisible as the hemostatic clip had blocked the blood flow (Fig. 4g-i, ii). Unfortunately, the fluorescent signal of Y6CT-NPs cannot be detected in the distal end of the right ureter after releasing the hemostatic forceps (Fig. 4g-iii), suggesting a dispersal of its blood supply function. After removal of the damaged ureter, an adequate blood supply can be acquired at the right ureter with Y6CT-NPs spilling out onto the gauze (Fig. 4g-iv). These results demonstrate that Y6CT-NPs can be used to evaluate the blood supply of the ureter of the transplanted kidney under a laparoscopic-light source, thus assisting surgeons to identify the active ureter and avoid necrosis of the end of the transplanted kidney ureter.

## Discussion

In summary, although imaging contrast agents are widely considered as one of the most promising clinical precision medicines, the imaging contrast agents reported so far are mainly activated by lasers or X-ray. In this study, we have developed white-light activatable NIR-II fluor-ophore by the nano-particliclization of Y6CT molecules. Owing to the strong intramolecular/intermolecular D-A interactions in aggregation state, the Y6CT-NPs exhibit ultrahigh QY × ε values of 13,315.1 upon the white-light activation that is two times higher than that of the brightest laser-activating contrast agent, thus can allow NIR-II imaging of the vascular structure and intraoperative monitoring of HIR with superb resolution and SNR under white-light illumination activation. In addi-tion, Y6CT-NPs can be directly applied in visible and precise identifi-cation of the renal vasculature, post-reconstruction evaluation of renal allograft vascular integrity, and analysis of the ureter's blood supply in a manner that is akin to real clinical scenarios. This work provided

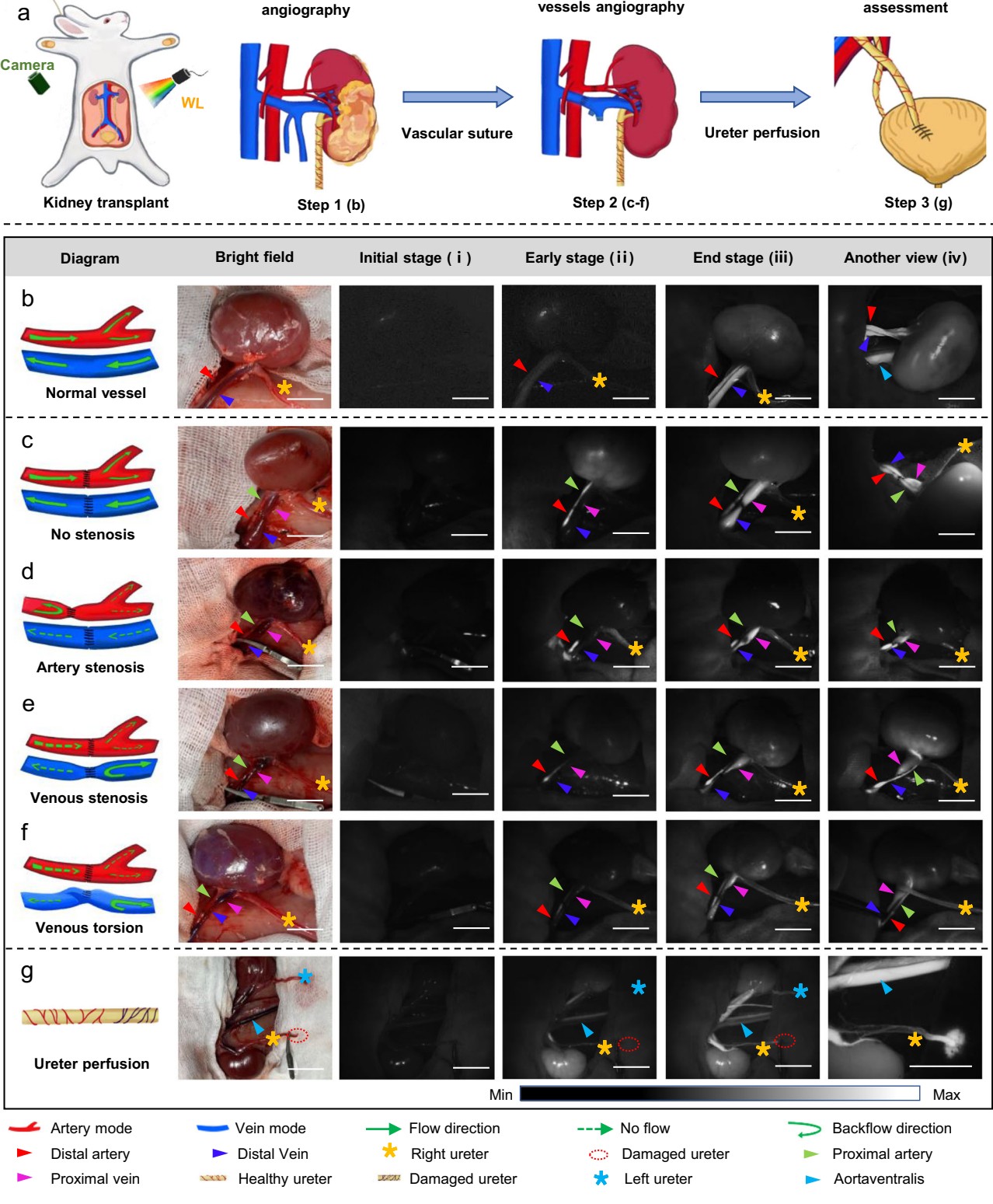

**Fig. 4 | White-light activated Y6CT-NPs for imaging-guided surgery of kidney transplant. a** Schematic diagram of the kidney transplant procedure. Camera: InGaAs camera, WL white-light (laparoscopic-light). **b** The diagram, bright-field image at initial stage, and fluorescence images at different stages of donor kidney vessels after intravenous injection of Y6CT-NPs (300 μM, 2 mL). The diagram, bright-field image at initial stage, and fluorescence images at different stages of kidney vessels in different types of vascular anastomotic abnormalities (no stenosis (**c**), artery stenosis (**d**), venous stenosis (**e**), venous torsion (**f**)) when kidney transplantation was processed. **g** The diagram, bright-field image at initial stage, and fluorescence images at different stages of ureter. The emission of Y6CT-NPs was collected in NIR-II windows and excited by a laparoscopic-light (20 mW cm⁻²). Scale bar: 1 cm.

molecular design guidelines in developing white-light activatable NIR-II contrast agents for precision medicine.

## Methods

All animals were cared for in accordance with the guidelines outlined in the Guide for the Care and Use of Laboratory Animals, and the procedures were approved by the Institutional Animal Care and Use Committee at the Inner Mongolia University (IMU-2022-mouse-047 and IMU-2023-rabbit-048). The animals were accommodated in an environment characterized by a 12 h light-dark cycle, with a consistent temperature of 25 °C and humidity level of 50%. Sex was not considered in the study design and analysis.

### In vivo angiography

Female BALB/c nude mice (6 weeks old and weighing 18–20 g) were purchased from Spiff Biotechnology Co., Ltd. (Beijing, China). Prior to the experiment, the mice were anesthetized via 2% isoflurane and then administered a tail-vein injection of 100 μL Y6CT-NPs or ICG (300 μM). Subsequently, the in vivo NIR-II fluorescence of the blood vessels in the abdomen was imaged using the NIR-II fluorescence imaging system with 900, 1000, and 1100 nm LP-filters upon white-light illumination activation.

### Monitoring hepatic ischemia reperfusion

The mice were anesthetized by intraperitoneal injection of a mixture of Ketamine (100 mg kg$^{-1}$) and Xylazine (10 mg kg$^{-1}$), followed by maintenance with 2% isoflurane prior to the experiment initiation. The abdominal skin was then cleaned and disinfected before the skin and muscle layers were cut in sequence. A 1 cm mid-abdominal incision was made, the abdominal cavity was opened, and the hepatic hilum of the left and middle lobes of the liver (portal vein and hepatic artery supplying blood to the liver in the left and middle lobes of the liver) was carefully detached. Noninvasive vascular clamps were used to clamp the portal vein and hepatic artery in the left lobe, and then a tail-vein injection of 100 μL Y6CT-NPs (300 μM) was administered. The liver region was imaged using the NIR-II fluorescence imaging system with white-light illumination activation. After 1 h, the hemostatic forceps were released and the liver was imaged again consecutively. Subsequently, 50 μL of Y6CT-NPs were injected and the liver was imaged with the NIR-II imaging system once more. After the experiment, the abdominal incision was sutured using polyglactin sutures (Ningbo Medical Needle Company, China).

### Imaging-guided surgery of kidney transplant

The male/female New Zealand rabbits (3 months old and weighing 1.5 - 2.0 kg) were obtained from National Institutes for Food and Drug Control (Beijing, China). For the forthcoming kidney transplantation experiments in rabbits, the concentrations of Y6CT-NPs utilized will be consistently set at 300 μM.

### Donor kidney vessels angiography

Before the experiment, male and female New Zealand rabbits were anesthetized with a mixture of Ketamine (30 mg kg$^{-1}$) and Xylazine (3 mg kg$^{-1}$) via intramuscular injection, followed by maintenance under 2.5% isoflurane. The abdominal skin was then cleaned and disinfected before the skin and muscle layers were incised in succession. The stomach and intestines were pushed to the right to expose the left renal area for NIR-II fluorescence imaging. Subsequently, 2 mL of Y6CT-NPs or ICG (300 μM) was administered to the rabbit via an intravenous needle (SHINVA Medical Instrument Company, China). The left renal region was then imaged using the NIR-II fluorescence imaging system with a laparoscopic LED light activation. The laparoscopic-light was placed 10 cm away from the vessels area and the emission filter was set to 1000 nm.

### Transplanted kidney vessels angiography

The left renal artery and vein were disconnected in the middle, and a vascular clamp was used to close the proximal end. 8-0 polyglactin sutures was then used to end-to-end suture the renal artery and vein with appropriate needle spacing. Subsequently, 2 mL of Y6CT-NPs solution was injected into the ear vein of the rabbit via a vein detained needle. Once the vascular clamp was released, NIR-II fluorescence images of vessels were collected in the rabbit under laparoscopic-light activation.

### Artery stenosis model

Disconnect the left renal artery and vein in the middle, then use a vascular clamp to occlude the proximal end of the left renal artery and vein. For the left renal vein, 8-0 polyglactin sutures is used for intermittent suturing, with appropriate needle spacing. For the left renal artery, an artery stenosis model is created by suturing one side of the sliding thread to the other side of the vascular wall. Subsequently, 2 mL of Y6CT-NPs was injected into the ear vein of the rabbit via a vein detained needle. Once the vascular clamp was released, NIR-II fluorescence images of vessels were collected in the rabbit under laparoscopic-light activation.

### Venous stenosis model

Disconnecting the left renal artery and vein in the middle, a vascular clamp was applied to the proximal ends to create an obstruction model of the transplanted renal vein. For the renal artery, 8-0 polyglactin sutures is used for intermittent suturing, with appropriate needle spacing. For the renal vein, a venous stenosis model was created by suturing one side of the sliding thread to the other side of the vascular wall. Subsequently, 2 mL of Y6CT-NPs was injected through a venous needle into the ear vein of the rabbit. Post-injection, NIR-II fluorescence images of the vessels were captured under laparoscopic-light activation.

### Venous torsion model

Disconnect the left renal artery and vein in the middle, then use a vascular clamp to close the proximal end of the left renal artery and vein, and then use 8-0 polyglactin sutures to sequentially end-to-end suture the renal artery and vein. Twist the left renal vein after anastomosis. Then 2 mL of Y6CT-NPs solution was injected into ear vein of rabbit via a vein detained needle. After releasing the vascular clamp immediately, NIR-II fluorescence images of the vessels were captured under laparoscopic-light activation.

### Ureter perfusion model

Disconnect the left renal artery and vein in the middle, then use a vascular clamp to close the proximal end of the left renal artery and vein, and then use 8-0 polyglactin sutures to sequentially end-to-end suture the renal artery and vein. Treat the right side in the same way. Cut off the distal ureters on both sides, and then use a vascular clamp to clamp the right ureteral end, with the vascular clamp located 1 cm away from the right ureteral end. Then 2 mL of Y6CT-NPs was injected into ear vein of rabbit via a vein detained needle. After releasing the vascular clamp, NIR-II fluorescence images of the vessels were captured under laparoscopic-light activation. Subsequently, the clamped ureter was cut out and subjected to overall and local magnification NIR-II imaging.

### Reporting summary

Further information on research design is available in the Nature Portfolio Reporting Summary linked to this article.

## Data availability

All original data generated in this study are provided in the Supplementary Information/Source Data file. The X-ray crystallographic

coordinates for structures reported in this study have been deposited at the Cambridge Crystallographic Data Centre (CCDC), under deposition number 2302102. These data can be obtained free of charge from The Cambridge Crystallographic Data Centre via www.ccdc.cam.ac.uk/data_request/cif. There are no restrictions on data availability in the current work. Besides, the full image dataset is available from the corresponding author upon request. Source data are provided with this paper.

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

## Acknowledgements

This study was supported by National Key Research and Development Program of China grant 2022YFA1205200 (L.W.), National Natural Science Foundation of China grant 22161034 (J.W.), National Natural Science Foundation of China grant 51721002 (L.W.), National Natural Science Foundation of China grant 22165020 (G.J.), National Natural Science Foundation of China grant 22371151 (J.W.), National Natural Science Foundation of China grant 52033003 (L.W.) and Grassland Talent Program of Inner Mongolia Autonomous Region of China grant 12000-12102807 (J.W.), Science and Technology Leading Talent Team in Inner Mongolia Autonomous Region grant 2022LJRC0001 (J.W.), Natural Science Foundation of Shandong Province ZR2021QH285 (J. D.).

## Author contributions

J.W., G.J. and L.W. initiated and supervised the project. C.L. and J.D. conceived the research and method. C.L., J.G., Y.Z. and M.Y. designed the simulation and experiment and conducted the experiments. C.L. and J.D. built the experimental system. C.L., J.W., G.J., L.W., B.T. and J.D. analyzed the results. C.L., J.W., L.W., G.J., B.T. and J.D. prepared the paper with input from all authors. All authors discussed the research.

## Competing interests

The authors declare no competing interests.
