## [Peer Review File · Nature Communications]

Reviewers' Comments:

Reviewer #1:

Remarks to the Author:

In this manuscript, Li and co-worker firstly reported a white-light activatable, planar A-D-A structured NIR-II organic nanomaterial, Y6CT-NPs, with high quantum yield and superior molar extinction coefficient. Y6CT-NPs exhibited high brightness with $QY \times \epsilon$ values of 13315.1, which is more than 2-times that of the brightest contrast agent previously reported. The exceptional performance of Y6CT-NPs enabled successful in vivo imaging of mouse abdominal vessels, monitoring of surgical procedures in a mouse model of hepatic ischemia reperfusion injury, and guidance of kidney transplantation in rabbits. The reviewer thinks that this is a very meaningful and challenging work in potential clinical applications. Overall, the manuscript is very interesting, innovative, well organized and holds solid evidence in terms of structural characterization, crystal and theoretical analysis, in vivo angiography and imaging-guided surgery process. The research contents of the manuscript are suitable for the scope of Nature Communication. Therefore, it can be accepted for publication in Nature Communication after minor revision.

- (1) Many largely π -conjugated NIR-II nanomaterials show low quantum yield in aqueous solution due to their strong π - π packing in the aggregated state. Why does Y6CT-NPs exhibit such a high fluorescence emission?
- (2) Can Y6-NPs be illuminated under white light activation? The emission spectrum of Y6-NPs under white light activation should be measured.
- (3) The fluorescence images of Y6-NPs under white light irradiation should be complemented in Figure 1g.
- (4) Why the diameters of Y6CT-NPs were so large (176 nm)? Do they have good stability under long term storage?
- (5) In general, many largely π -conjugated NIR-II molecules should have a good thermal property under laser irradiation, which is unfavorable for in vivo imaging. Thus, how about the photothermal properties of Y6CT-NPs under white light irradiation?
- (6) The evaluation of biological safety of new system is very important. The evaluation methods presented in this work are not specific enough. The hemocompatibility of Y6CT-NPs should be performed.
- (7) "Y6CT-NPs exhibited an impressive QY of 16.12% in aqueous solution" The excitation sources used in the QY tests should be mentioned.
- (8) What is the short-term metabolic pathway of Y6CT-NPs? Ex vivo biodistribution studies of mice should be conducted at 1st, 3rd, 5th, and 7th day post-injection of Y6CT-NPs.

Reviewer #2:

Remarks to the Author:

This work reports Y6CT nanoparticles (Y6CT-NPs) that can emit NIR-II fluorescence under visible light excitation and uses the Y6CT-NPs as the contrast agent for in vivo imaging of hepatic ischemia reperfusion and for real-time monitoring of kidney transplantation surgery. The use of white light sources, such as a laparoscopic LED, as the excitation sources in NIR-II imaging can make the imaging be more easily and conveniently carried out compared with the laser or X-ray excitation. Therefore, the content is suitable for the consideration of publication in Nature Communications. However, the following questions need to be addressed.

- (1) The authors stated that "all the reported NIR-II imaging contrast agents are activated by expensive high-power lasers or X-ray" and therefore claimed that the Y6CT-NPs are "the first white-light activatable NIR-II imaging agent". These statement and claim are incorrect. While the Y6CT-NPs may be the first white-light activatable organic NIR-II imaging agent, white light sources, such as white LED bulbs and white LED flashlights, were recently used by Chen et al. to excite inorganic CaTiO₃:Pr³⁺ imaging agent for persistent luminescence-based NIR-II imaging,

see Chemical Engineering Journal 446 (2022) 137473
(<https://doi.org/10.1016/j.cej.2022.137473>).

(2) On page 5, paragraph 1, the authors claimed that “Y6CT exhibits strong light-harvesting capability at 500-1000 nm both in solution and in nanoparticle state (Fig. 1c)”; however, in Fig. 2c, the absorption cut-off wavelength for Y6CT in solutions (in CHCl₃ and THF) is clearly shown at around 800 nm. The absorption band for Y6CT-NPs was extended to about 1000 nm.

(3) The Y6CT-NPs exhibit much broader and large redshift absorption band (Fig. 1c) and large redshift fluorescence emission band (Fig. 1d) compared with Y6CT solution state, for which the authors attributed them to efficient molecular stacking in aggregate state. Since the spectral property of a material is inherent and the reported broadening and shifts are very large for the same material (Y6CT) in different states, more comprehensive explanation is needed. Also to Fig. 1d, the authors claimed the “Y6CT-NPs present a bright NIR-II emission extending to 1400 nm”, Fig. 1d actually shows that the NIR-II emissions of Y6CT in solutions at ~1250-1400 nm are even higher than Y6CT-NPs, making them seemly more suitable for NIR-II imaging.

(4) The QY of Y6CT-NPs in aqueous solution was determined to be as high as 16.12% using NIR-II fluorescent IR26 dye (QY = 0.5%) as a reference. Fig. S4 shows that the emission of IR26 is mostly in the NIR-II range (>1000 nm)(the PL intensity axis of Fig. S4a was set so high that the emission spectra of IR26 cannot be seen at all), however, the wavelength range of Y6CT-NPs used for calculation is 850 – 1400 nm, which includes not only the NIR-II range but also the NIR-I range and the NIR-I range (850-1000 nm) covers a bigger portion of the emission spectrum. Therefore, the value of the QY of Y6CT-NPs needs to be reconsidered. The resulting NIR-II fluorescence brightness (13315.1) of Y6CT-NPs needs to be reconsidered too. Actually, the value of 13315.1, a selling point by the authors, cannot give the readers (including the reviewer) a meaningful sense about the brightness of the material.

(5) To show the superiority of Y6CT-NPs, the authors compared the Y6CT-NPs with many other reported organic NIR-II fluorophores, such as the ones in Fig. 1f (as well as the description in page 5) and many in Table S1. If the measurements were not conducted at the exactly same experimental conditions, such a comparison among different materials measured by different groups does not mean much and is even inappropriate. Particularly considering the questionable QY and brightness raised in Question #4.

(6) The NIR-II imager used in this work is a Series III 900/1700 In Vivo Imaging System manufactured by Suzhou NIR-Optics Co., Ltd., China. However, the reviewer cannot find related information by searching this imager and the company. In NIR-II imaging systems, the detectors are usually InGaAs focal plane array. However, in Fig. 3c and Fig. 4a, it indicated CCD (charge coupled device).

(7) Throughout the manuscript, the authors repeatedly claimed the potential of their Y6CT-NPs in practical clinical surgery applications. While there is certain potential, it is far far away. Therefore, the authors may consider to lower the tone a bit.

(8) In Supplementary Information (SI), page S2, paragraph 2, the writing is somewhat random and lacks of logic. The bottom part on page S9 was not correctly displayed.

(9) The reviewer feels the lack of certain connection between the main text and the SI. Without intentionally reading the SI, the main text cannot be understood clearly and correctly, particularly all the experiments were given in SI. Therefore, some basic information about experiment needs to be mentioned in the main text.

Reviewer #3:

Remarks to the Author:

In this manuscript, the authors developed NIR-II nanoparticles by the nanoparticlization of Y6CT molecules and DSPE-PEG and employed such nanomaterials for imaging-guided surgery, including imaging of hepatic ischemia reperfusion and monitoring of kidney transplantation surgery. Indeed, the fluorophore used in this study has recently been published on Adv. Mater. 35, e2208229 (2023), which largely reduced the novelty to this work. In addition, the authors used white light to excite the fluorophore for imaging-guided surgery, however, the channel of naked eyes and image-

guided surgery are independently used in practice. Thus, white light excitation is of less significance. Moreover, this work has some critical problems in terms of poor characterization data and lack of control experiments in imaging studies. Also, the imaging performance is far from impressive. Thus, I don't recommend its publication in this top tier journal.

Below are some suggestions that the authors might consider to improve this work.

1. In Supplementary Table 1, the quantum yield of Y6CT-NPS was calculated to be 16.12% based on the quantum yield of ICG in water (1%) as reference. However, most literatures reported that the quantum yield of ICG in water is $\sim 2.5\%$. The authors should double-check the characterization data presented in this article.
2. In both animal experiments, the authors did not provide sufficient control experiments for comparison. Such comparisons studies should involve either other fluorophores such as ICG and IR26 or other excitation methods (power laser).
3. In Figure 3, the signal-to-noise ratio of the imaging is only 1.6. Such a low SNR is difficult to meet the requirement for imaging-guided surgery. In addition, the imaging performance in Figure 4 is poor under the white light excitation, thus really difficult to distinguish the vessels.
4. The authors are suggested to test the luminescence efficiency as well as the quantum yield of Y6CT-NPS in other relevant solvents such as FBS and DCE.
5. The fluorophore employed in this study has recently been published on Adv. Mater. 35, e2208229 (2023), which largely reduced the novelty to this work.

Point-by-point response to the Reviewers' comments:

Reviewer #1

In this manuscript, Li and co-worker firstly reported a white-light activatable, planar A-D-A structured NIR-II organic nanomaterial, Y6CT-NPs, with high quantum yield and superior molar extinction coefficient. Y6CT-NPs exhibited high brightness with $QY \times \epsilon$ values of 13315.1, which is more than 2-times that of the brightest contrast agent previously reported. The exceptional performance of Y6CT-NPs enabled successful in vivo imaging of mouse abdominal vessels, monitoring of surgical procedures in a mouse model of hepatic ischemia reperfusion injury, and guidance of kidney transplantation in rabbits. The reviewer thinks that this is a very meaningful and challenging work in potential clinical applications. Overall, the manuscript is very interesting, innovative, well organized and holds solid evidence in terms of structural characterization, crystal and theoretical analysis, in vivo angiography and imaging-guided surgery process. The research contents of the manuscript are suitable for the scope of Nature Communication. Therefore, it can be accepted for publication in Nature Communication after minor revision.

Response: We sincerely thank the Reviewer for his or her positive comments and professional and constructive suggestions, which will further improve the quality of our manuscript.

(1) Many largely π -conjugated NIR-II nanomaterials show low quantum yield in aqueous solution due to their strong π - π packing in the aggregated state. Why does Y6CT-NPs exhibit such a high fluorescence emission?

Response: It is noted that many fluorophores exhibit *H*-type or disordered aggregation, resulting in diminished NIR absorption and fluorescence efficiency upon NPs formation. In contrast, our Y6CT crystals demonstrate a *J*-aggregation stacking mode and strong intermolecular D-A interactions, which can facilitate constructive coupling of excited-state transition dipoles. This unique characteristic of Y6CT leads to red-

shifted absorption/emission wavelengths, elevated molecular extinction coefficients (ϵ) and increased quantum yields (QY) in the aggregated state. Consequently, Y6CT-NPs exhibit high NIR-II brightness ($QY \times \epsilon$).

(2) Can FY6-NPs be illuminated under white light activation? The emission spectrum of FY6-NPs under white light activation should be measured.

Response: According to your suggestion, fluorescence spectra of FY6-NPs under white-light activation has been performed. According to results in Supplementary Fig. 6, the PL intensity of FY6-NPs at the maximum emission wavelength is only 0.4-times that of Y6CT-NPs. Related discussion has been added in blue words on Page 4 (line 14, left column) and the added Fig. S6 on Page S8.

On Page 4 in the revised manuscript:

In addition, the fluorescence spectra of FY6-NPs, as a nanomaterial outlined in our previous study³⁹, was investigated under white-light activation. Although FY6-NPs demonstrate NIR-II emission upon white-light excitation, their PL intensity at maximum emission wavelength is only 0.4-times that of Y6CT-NPs (Supplementary Fig. 6). These results afford the Y6CT-NPs a probability for NIR-II imaging in biological systems under white-light activation.

On Page S8 in the revised Supporting Information:

Supplementary Fig. 6. Fluorescence spectra of FY6-NPs and Y6CT-NPs ($E_x =$ Laparoscopic-light, 643 nm LP filter).

(3) The fluorescence images of FY6-NPs under white light irradiation should be complemented in Figure 1g.

Response: According to your suggestion, we have added NIR-II fluorescent images of FY6-NPs under white-light illumination and corresponding daylight photographs in Fig. 1g. Moreover, an analysis of the NIR-II fluorescence signal intensity of FY6-NPs also has been illustrated in Fig. 1g. Related discussion has been added in blue words on Page 4-6.

On Page 4-6 in the revised manuscript:

“their NIR-II imaging property was compared with that of FY6-NPs and several commercial laser-activated NIR-II dyes,”

“Very impressively, Y6CT-NPs exhibit a bright NIR-II emission, with an intensity of over 25-times higher than that of commercial dyes and nearly 4-times stronger than that of FY6-NPs, suggesting the great potential for *in vivo* imaging under white-light activation.”

(4) Why the diameters of Y6CT-NPs were so large (176 nm)? Do they have good stability under long term storage?

Response: We inferred that the diameters of Y6CT-NPs may be related with both the molecular structure and the hydrophobicity/hydrophilicity of the compounds. Various researchers also reported NPs with diameters similar as that of Y6CT-NPs, which exhibited good stability under long term storage (*Biomaterials* **243**, 119934 (2020); *Chem. Eng. J.* **449**, 137846 (2022); *ACS Nano* **16**, 3323 (2022); *Adv. Funct. Mater.* **30**, 1910301 (2020)). Moreover, as recommended, we assessed the stability of Y6CT-NPs under long term storage of 14 days. Data in Supplementary Fig. 4 showed that the hydrodynamic diameter of Y6CT-NPs remained largely unchanged during 14 days' storage, indicating their good stability in PBS solution. Results and related discussion have been added in blue words on Page 3 (line 7, right column) and the added Fig. S4 on Page S8.

On Page 3 in the revised manuscript:

Additionally, the stability of Y6CT-NPs in long-term storage was further analyzed. As shown in Supplementary Fig. 4, the diameter of Y6CT-NPs experienced negligible variations over a 14-day period in a PBS solution, indicating a favorable stability.

On Page S8 in the revised Supporting Information:

Supplementary Fig. 4. The diameter changes of Y6CT-NPs in PBS within 14 days (n = 3).

(5) In general, many largely π -conjugated NIR-II molecules should have a good thermal property under laser irradiation, which is unfavorable for in vivo imaging. Thus, how about the photothermal properties of Y6CT-NPs under white light irradiation?

Response: According to your suggestion, we investigated the photothermal efficiency of Y6CT-NPs under white-light. As shown in the added Supplementary Fig. 11, Y6CT-NPs displayed negligible temperature variations (< 2 °C) under continuous white-light activation. This finding provides additional evidence supporting the excellent safety of Y6CT-NPs under white-light activation for NIR-II imaging in our manuscript. Please see the blue words on Page 6 (line 23, left column) and Page S3 and the added Fig. S11 on Page S10.

On Page 6 in the revised manuscript:

Furthermore, to exclude the unfavorable photothermal injury, the photothermal efficiency of Y6CT-NPs was assessed under white-light activation. As depicted in Supplementary Fig. 11, the temperature variation of Y6CT-NPs remained consistently

stable without notable increase under continuous exposure for 6 min, even at high concentrations of 100 μM , emphasizing favorable safety profile as a contrast agent for imaging purposes.

On Page S3 in revised Supporting Information:

Photothermal properties of Y6CT-NPs. The white-light power density was calibrated with a spectro-densitometer to ensure accuracy. The temperature curves of Y6CT-NPs (200 μL) in water were recorded at various concentrations (0, 20, 50, 100 μM) upon white-light exposure (20 mW cm^{-2}) for 6 min.

On Page S10 in the revised Supporting Information:

Supplementary Fig. 11. Photothermal curves of Y6CT-NPs with different concentrations under white-light excitation.

(6) The evaluation of biological safety of new system is very important. The evaluation methods presented in this work are not specific enough. The hemocompatibility of Y6CT-NPs should be performed.

Response: Thank you for your constructive suggestion. According to your suggestion, we conducted hemocompatibility test of Y6CT-NPs. PBS and 1% Triton X-100 (Triton) were used as negative control and positive control, respectively. As shown in Supplementary Fig. 13, the incubation of blood with Y6CT-NPs for 2 h resulted in a hemolytic ratio of less than 1% for all the tested concentrations. According to ISO/TR 7406, samples with hemolytic rates less than 5% (the critical safe hemolytic ratio for biomaterials) are considered nonhemolytic, which suggest that Y6CT-NPs possess good

blood compatibility. Please see the added discussion in blue words on Pages 8 (line 13, left column) and S4, as well as the added Fig. S13 on Page S11.

On Page 8 in the revised manuscript:

In addition, systemic toxicity evaluation revealed that Y6CT-NPs had negligible organ and hematological toxicity *in vivo* (Supplementary Fig. 13 and Fig. 14, Supplementary Table 3 and Table 4).

On Page S4 in in the revised Supporting Information:

Hemocompatibility test. Whole blood was collected from healthy mice, centrifuged at 2000 rpm for 10 min to separate the serum and blood cells. The blood cells were then washed thrice with PBS. Subsequently, 50 μ L of blood cells were treated with varying concentrations of Y6CT-NPs (1, 2, 5, 10, 20, 30, 100 μ M) in polyethylene pipe containing 1 mL of PBS, and incubated for 2 h at 37 $^{\circ}$ C. Phosphate buffer saline buffer (PBS) and 1% Triton X-100 (Triton) were used as negative control and positive control, respectively. Then, all samples were centrifuged for 10 min, and images were captured. The absorbance of the supernatant was measured at 540 nm. To correct for compound absorbance, a Y6CT-NPs in PBS solution with the same concentration was used as the blank group. Hemolysis was calculated using the following formula:

$$\text{Hemolysis (\%)} = \frac{A_{\text{Blood}} - A_{\text{Sample}}}{A_{\text{Triton}} - A_{\text{PBS}}} \times 100\% \quad (\text{c})$$

where A_{Blood} and A_{Sample} represent the absorbance of blood-containing and blood-free PBS solution treated with varying concentrations of Y6CT-NPs, respectively. A_{Triton} and A_{PBS} denote the absorbance of blood-containing PBS solution treated with and without Triton X-100, respectively.

On Page S11 in the revised Supporting Information:

Supplementary Fig. 13. The hemocompatibility of Y6CT-NPs. Hemolytic percentage of red blood cells after treatment with Y6CT-NPs at various concentrations ranging from 1 to 100 μM after 2 h incubation. PBS and 1% Triton X-100 (Triton) were used as negative control and positive control, respectively (n = 3).

(7) "Y6CT-NPs exhibited an impressive QY of 16.12% in aqueous solution" The excitation sources used in the QY tests should be mentioned.

Response: According to your suggestion, we have revised this section to "As shown in Fig. 1e and Supplementary Fig. 7, Y6CT-NPs display an impressive QY of 16.12% in the 850-1400 nm range in aqueous solution (QY of 8.62% in the 1000-1400 nm range) using 808 nm laser as excitation source, which is ~ 32 -times that of IR26 and ~ 4 -times higher than our previously reported FY6-NPs (QY of 4.08% and 2.24% in the spectral ranges of 850-1000 nm and 1000-1400 nm, respectively)³⁹." Please see the blue words on Page 4.

On Page 4 of the revised manuscript:

To evaluate the luminescence efficiency of Y6CT-NPs, their QY was determined using commercially available NIR-II contrast agent 4-[2-[2-chloro-3-[2-(2-phenyl-2H-thiochromen-4-yl)ethenyl]cyclohex-2-en-1-ylidene]ethylidene]-2-phenylthiochromene (IR26) (QY = 0.5%) in 1,2-dichloroethane (DCE) as the reference^{24,39}. As shown in Fig. 1e and Supplementary Fig. 7, Y6CT-NPs display an impressive QY of 16.12% in the 850-1400 nm range in aqueous solution (QY of 8.62% in the 1000-1400 nm range) using 808 nm laser as excitation source, which is ~ 32 -

times that of IR26 and ~ 4-times higher than our previously reported FY6-NPs (QY of 4.08% and 2.24% in the spectral ranges of 850-1400 nm and 1000-1400 nm, respectively)³⁹.

(8) *What is the short-term metabolic pathway of Y6CT-NPs? Ex vivo biodistribution studies of mice should be conducted at 1st, 3rd, 5th, and 7th day post-injection of Y6CT-NPs.*

Response: According to your suggestion, *ex vivo* biodistribution study of mice was conducted at 1st, 3rd, 5th, and 7th days post-injection of Y6CT-NPs. The experimental results suggest that the liver and spleen function as the main metabolic organs for Y6CT-NPs within the organism (Supplementary Fig. 18). Please see the blue words on Pages 8, S4 and the added Fig. S18 on Page S13.

On Page 8 of the revised manuscript:

In addition, the short-term metabolic behavior of Y6CT *in vivo* was assessed by conducting *ex vivo* biodistribution studies in mice on the 1st, 3rd, 5th, and 7th days post-administration of Y6CT-NPs (Supplementary Fig. 18). The results revealed bright NIR-II fluorescence signals in the liver and spleen, indicating the principal metabolic pathways of Y6CT-NPs through these organs. A subsequent gradual decrease of fluorescence intensity in these two organs suggested the progressive metabolism of Y6CT-NPs *via* the liver and spleen.

On Page S4 in the revised Supporting Information:

Metabolic pathway of Y6CT-NPs. BALB/c nude mice were intravenously administered Y6CT-NPs (300 μ M, 100 μ L), followed by euthanasia on the 1st, 3rd, 5th, and 7th days post-injection. Subsequently, the major organs (heart, liver, spleen, lung, kidney) of these mice were excised, and their fluorescent images were acquired utilizing the NIR-II fluorescence imaging system with 1000 nm LP-filters upon activation by white-light illumination.

On Page S13 in the revised Supporting Information:

Supplementary Fig. 18. The photographs and NIR-II imaging of major organs (heart, liver, spleen, lung, kidney) at 1st, 3rd, 5th, and 7th days post-injection of Y6CT-NPs. E_x = White-light illumination.

Reviewer #2

This work reports Y6CT nanoparticles (Y6CT-NPs) that can emit NIR-II fluorescence under visible light excitation and uses the Y6CT-NPs as the contrast agent for in vivo imaging of hepatic ischemia reperfusion and for real-time monitoring of kidney transplantation surgery. The use of white light sources, such as a laparoscopic LED, as the excitation sources in NIR-II imaging can make the imaging be more easily and conveniently carried out compared with the laser or X-ray excitation. Therefore, the content is suitable for the consideration of publication in Nature Communications. However, the following questions need to be addressed.

Response: We sincerely thank the Reviewer for his or her positive and valuable comments, and try our best to revise this manuscript according to these suggestions as follows:

(1) The authors stated that “all the reported NIR-II imaging contrast agents are activated by expensive high-power lasers or X-ray” and therefore claimed that the Y6CT-NPs are “the first white-light activatable NIR-II imaging agent”. These statement and claim are incorrect. While the Y6CT-NPs may be the first white-light activatable organic NIR-II imaging agent, white light sources, such as white LED bulbs and white LED flashlights, were recently used by Chen et al. to excite inorganic CaTiO₃:Pr³⁺ imaging agent for persistent luminescence-based NIR-II imaging, see Chemical Engineering Journal 446 (2022) 137473 (<https://doi.org/10.1016/j.cej.2022.137473>).

Response: Thank you very much for your helpful comments. We have revised our statements. Please see the blue words on Pages 1 and 2.

On Page 1 of the revised manuscript:

While second near-infrared (NIR-II) fluorescence imaging is a promising tool for real-time surveillance of surgical operations, the previously reported organic NIR-II luminescent materials for in vivo imaging are predominantly activated by expensive lasers or X-ray with high power and poor illumination homogeneity, which significantly

limits their clinical applications. Herein, we report the first white-light activatable NIR-II organic imaging agent by taking advantages of the strong intramolecular/intermolecular D-A interactions of conjugated Y6CT molecules in nanoparticles (Y6CT-NPs), with the brightness of as high as 13315.1, which is over two times that of the brightest laser activated NIR-II organic contrast agents reported thus far.

On Page 2 of the revised manuscript:

“However, the previously reported organic NIR-II imaging contrast agents are mainly activated by expensive high-power lasers or X-ray¹⁸⁻²⁴.”

“Although some small organic NIR-II fluorophores including D- π -A, D-A-D, and A-D-A type have been developed for imaging-guided surgery in the past few years^{6,30,34-39}, they were majorly activated upon laser or X-ray due to their low-absorptivity in the visible region.”

(2) On page 5, paragraph 1, the authors claimed that “Y6CT exhibits strong light-harvesting capability at 500-1000 nm both in solution and in nanoparticle state (Fig. 1c)”; however, in Fig. 2c, the absorption cut-off wavelength for Y6CT in solutions (in CHCl₃ and THF) is clearly shown at around 800 nm. The absorption band for Y6CT-NPs was extended to about 1000 nm.

Response: We are deeply sorry for our carelessness. We have revised the expression, and the corresponding statements have been revised in blue words on Page 3.

On Page 3 of the revised manuscript:

From the UV-*vis*-NIR and photoluminescence (PL) spectroscopy, Y6CT exhibits strong absorption at 500-800 nm in CHCl₃ and THF solution. And notably, in nanoparticle state, Y6CT-NPs maintained excellent light-harvesting capability with broad absorption extended to 1000 nm (Fig. 1c).

(3) The Y6CT-NPs exhibit much broader and large redshift absorption band (Fig. 1c) and large redshift fluorescence emission band (Fig. 1d) compared with Y6CT solution

state, for which the authors attributed them to efficient molecular stacking in aggregate state. Since the spectral property of a material is inherent and the reported broadening and shifts are very large for the same material (Y6CT) in different states, more comprehensive explanation is needed. Also to Fig. 1d, the authors claimed the “Y6CT-NPs present a bright NIR-II emission extending to 1400 nm”, Fig. 1d actually shows that the NIR-II emissions of Y6CT in solutions at ~1250-1400 nm are even higher than Y6CT-NPs, making them seemly more suitable for NIR-II imaging.

Response: Thanks for your valuable suggestions and comments. Spectra broadening and red shifts are commonly observed in organic molecules with largely π -conjugated structures, which is usually attributed to the formation of strong intermolecular interactions in the aggregated or solid states (*Nat. Commun.* **12**, 2376 (2021); *Adv. Mater.* **32**, 2003471 (2020); *Angew. Chem. Int. Ed.* **135**, e202303476 (2023)). Moreover, to further prove this point, we have investigated the absorption and emission spectra of Y6CT in mixed solutions under varying THF/H₂O ratios. Y6CT is soluble in THF, while is poorly soluble in water. Our results in Supplementary Fig. 5 indicate that an increase in water content triggered the transition of Y6CT from a monomeric state to aggregates, resulting in a gradual broadening and red shift of the absorption band, along with a red shift in the emission band (Supplementary Fig. 5). As for the elevation of base line in NIR-II emission (Fig. 1d) of Y6CT in CH₃Cl and THF solution, we inferred this may be due to the interference from the solvent or instrument. Thus we have re-scanned the NIR-II emission spectra of Y6CT in both solution and nanoparticle forms. The updated Fig. 1d shows that Y6CT-NPs exhibit significantly enhanced NIR-II fluorescence emission compared to that in solution. Data and related discussion have been added in the revised Supporting Information and revised manuscript, respectively. Please see the blue words on Pages 3, 4 and the added Fig. S5 on Page S8.

On Page 3-4 of the revised manuscript:

Particularly, Y6CT-NPs present a bright NIR-II emission extending to 1400 nm with two peaks of 947 nm and 1030 nm in aqueous solution, significantly redshifted compared to Y6CT solution state. In order to elucidate this phenomenon, absorption

and emission spectra of Y6CT in mixed solution with varied THF/water ratios were recorded. As shown in Supplementary Fig. 5, Y6CT demonstrated a redshift in the absorption peak from 706 to 789 nm and a widening of the absorption band with increasing water fractions. Likewise, the emission peak was also redshifted from 790 to 1034 nm. Importantly, the spectra of Y6CT in the aggregated state closely resemble those of the nanoparticle state, highlighting that the redshift and broadening of Y6CT-NPs' spectra primarily depend on efficient molecular stacking. In addition, the fluorescence spectra of FY6-NPs, as a nanomaterial outlined in our previous study³⁹, was investigated under white-light activation. Although FY6-NPs demonstrate NIR-II emission upon white-light excitation, their PL intensity at maximum emission wavelength is only 0.4-times that of Y6CT-NPs (Supplementary Fig. 6).

On Page S8 in the revised Supporting Information:

Supplementary Fig. 5. a-b The normalized UV-*vis*-NIR spectra (a) and PL spectra (b) of Y6CT in a mixture of THF and water with varying water fractions (f_w).

(4) The QY of Y6CT-NPs in aqueous solution was determined to be as high as 16.12% using NIR-II fluorescent IR26 dye (QY = 0.5%) as a reference. Fig. S4 shows that the emission of IR26 is mostly in the NIR-II range (>1000 nm) (the PL intensity axis of Fig. S4a was set so high that the emission spectra of IR26 cannot be seen at all), however, the wavelength range of Y6CT-NPs used for calculation is 850 – 1400 nm, which includes not only the NIR-II range but also the NIR-I range and the NIR-I range (850-1000 nm) covers a bigger portion of the emission spectrum. Therefore, the value of the QY of Y6CT-NPs needs to be reconsidered. The resulting NIR-II fluorescence brightness

(13315.1) of Y6CT-NPs needs to be reconsidered too. Actually, the value of 13315.1, a selling point by the authors, cannot give the readers (including the reviewer) a meaningful sense about the brightness of the material.

Response: Thank you very much for your professional suggestions. According to your suggestion, we determined the QY within the 1000-1400 nm range by employing IR26 as a reference, as illustrated in Supplementary Fig. 4 (now Supplementary Fig. 7). Our experimental findings reveal a QY of 8.62% for Y6CT-NPs, with a calculated NIR-II brightness of 7120.1, significantly surpassing that of previously reported NIR-II organic nanomaterials. In addition, we also adjusted the PL intensity axis in the emission spectra of IR26 to a suitable range and re-displayed it in Supplementary Fig. 4 (now Supplementary Fig. 7). Please see the blue words on Page 4, the added Fig. S7 on Page S9, and the updated Table S1 on Page S16.

On Page 4 of the revised manuscript:

“To evaluate the luminescence efficiency of Y6CT-NPs, their QY was determined using commercially available NIR-II contrast agent 4-[2-[2-chloro-3-[2-(2-phenyl-2H-thiochromen-4-yl)ethenyl]cyclohex-2-en-1-ylidene]ethylidene]-2-phenylthiochromene (IR26) (QY = 0.5%) in 1,2-dichloroethane (DCE) as the reference^{24,39}. As shown in Fig. 1e and Supplementary Fig. 7, Y6CT-NPs display an impressive QY of 16.12% in the 850-1400 nm range in aqueous solution (QY of 8.62% in the 1000-1400 nm range) using 808 nm laser as excitation source, which is ~ 32-times that of IR26 and ~ 4-times higher than our previously reported FY6-NPs (QY of 4.08% and 2.24% in the spectral ranges of 850-1400 nm and 1000-1400 nm, respectively)³⁹.”

“More importantly, Y6CT-NPs show excellent emission with a NIR-II brightness values (> 1000 nm) of 7120.1, far exceeding those of previously reported NIR-II organic small molecule nanomaterials (Fig. 1f and Supplementary Table 1).”

On Page S9 in the revised Supporting Information:

Supplementary Fig. 7. a-c The fluorescence spectra of IR26 in DCE (a), FY6-NPs (b) and Y6CT-NPs in deionized water (c) with different optical densities (ODs) under 808 nm laser excitation. **d** The linear fitting of the integrated PL intensity in the range of 1000-1400 nm vs. the absorbance values of Y6CT-NPs, FY6-NPs and IR26.

On Page S16 in the revised Supporting Information:

Supplementary Table 1. Photophysical characterization of previously reported organic NIR-II fluorophores³⁻²¹.

Name	Excitation light	ϵ_{\max} (L mol ⁻¹ cm ⁻¹)	λ_{em} (nm)	QY ^a (%)	Brightness (QY × ϵ)	Ref.
Y6CT-NPs	White-light	8.26 × 10⁴	947, 1030	16.12 8.62 (NIR-II)	13315.1 7120.1 (NIR-II)	This work
FY6-NPs	808 nm laser	7.64 × 10 ⁴	947, 1052	4.20	3208.8	3
BNDI-Me NPs	808 nm laser	16.4 × 10 ⁴	1104	1.40	2296.0	4
CPTIC NFs	808 nm laser	14.5 × 10 ⁴	1110	3.90	5655.0	5
BMIC-BO-4Cl NPs	880 nm laser	9.30 × 10 ⁴	1010	2.29	2129.7	6
IR-FFCHP NPs	808 nm laser	~ 1.25 × 10 ⁴	1038	7.30	912.5	7
L1013 NPs	808 nm laser	1.39 × 10 ⁴	1013	9.9	1376.1	8
TT3-oCB NPs	793 nm laser	2.07 × 10 ⁴	1062	4.60	952.2	9
TPE-BBT NPs	808 nm laser	1.40 × 10 ⁴ (DMSO)	955	31.5	4410	10
TTQiT NPs	808 nm laser	3.89 × 10 ⁴	1102	3.70	1439.3	11
TA1 NPs	808 nm laser	2.14 × 10 ⁴	893	0.08 ^b	17.12	12
HL3 dots	808 nm laser	0.93 × 10 ⁴	1125	11.70	1088.1	13
FD-1080 J-aggregate	1064 nm laser	5.00 × 10 ⁴	1370	0.54	270	14
ICG	808 nm laser	12.0 × 10 ⁴	822	1	1200	15
5H5	1064 nm laser	3.42 × 10 ⁴ (MeCN)	1069	2.6	889.2	16
CX-3	808 nm laser	5.13 × 10 ⁴ (DMSO)	1135	0.82	420.7	17
TPA-Et	808 nm laser	5.31 × 10 ⁴ (CH ₂ Cl ₂)	935	0.04 ^c	21.24	18
MB	623 nm LED	7.12 × 10 ⁴	ND	0.20	142.4	19
BAF4 NPs	1064 nm laser	~ 0.75 × 10 ⁴	ND	ND	ND	20
CCNU-1060 NPs	808 nm laser	1.60 × 10 ⁴	1065	0.3	48	21

(a) The QY was re-calculated using QY of IR26 = 0.5% in DCE as a standard. (b) ICG (NIR-II QY = 1%) in water were used as the reference for calculating the fluorescence QY of TA1 NPs. (c) Absolute QY.

(5) To show the superiority of Y6CT-NPs, the authors compared the Y6CT-NPs with many other reported organic NIR-II fluorophores, such as the ones in Fig. 1 (as well as the description in page 5) and many in Table S1. If the measurements were not conducted at the exactly same experimental conditions, such a comparison among different materials measured by different groups does not mean much and is even inappropriate. Particularly considering the questionable QY and brightness raised in Question #4.

Response: We agree that this comparison is indeed not quite appropriate, due to the fact that different materials cannot be measured at the exactly same experimental conditions. However, considering that there is a lack of uniform standard and an accurate method to offer an exact comparison by far, we have to do this comparison as others did, in order to give the audience a more clearly demonstration of the advantages of Y6CT-NPs. According to your suggestion, we also performed further calculation of the QY and brightness of Y6CT-NPs in the NIR-II region of 1000-1400 nm. Y6CT-NPs exhibited an NIR-II QY of 8.62% and brightness of 7120.1, which again evidenced the superiority. Please see the blue words on Page 4, the added Fig. S7 on Page S9 and the updated Table S1 on Page S16.

On Page 4 of the revised manuscript:

“To evaluate the luminescence efficiency of Y6CT-NPs, their QY was determined using commercially available NIR-II contrast agent 4-[2-[2-chloro-3-[2-(2-phenyl-2H-thiochromen-4-yl)ethenyl]cyclohex-2-en-1-ylidene]ethylidene]-2-phenylthiochromene (IR26) (QY = 0.5%) in 1,2-dichloroethane (DCE) as the reference^{24,39}. As shown in Fig. 1e and Supplementary Fig. 7, Y6CT-NPs display an impressive QY of 16.12% in the 850-1400 nm range in aqueous solution (QY of 8.62% in the 1000-1400 nm range) using 808 nm laser as excitation source, which is ~ 32-times that of IR26 and ~ 4-times higher than our previously reported FY6-NPs (QY of 4.08% and 2.24% in the spectral ranges of 850-1400 nm and 1000-1400 nm, respectively)³⁹.”

“More importantly, Y6CT-NPs show excellent emission with a NIR-II brightness values (> 1000 nm) of 7120.1, far exceeding those of previously reported NIR-II organic small molecule nanomaterials (Fig. 1f and Supplementary Table 1).”

On Page S9 in the revised Supporting Information:

Supplementary Fig. 7. a-c The fluorescence spectra of IR26 in DCE (a), FY6-NPs (b) and Y6CT-NPs in deionized water (c) with different optical densities (ODs) under 808 nm laser excitation. **d** The linear fitting of the integrated PL intensity in the range of 1000-1400 nm vs. the absorbance values of Y6CT-NPs, FY6-NPs and IR26.

(6) The NIR-II imager used in this work is a Series III 900/1700 *In Vivo* Imaging System manufactured by Suzhou NIR-Optics Co., Ltd., China. However, the reviewer cannot find related information by searching this imager and the company. In NIR-II imaging systems, the detectors are usually InGaAs focal plane array. However, in Fig. 3c and Fig. 4a, it indicated CCD (charge coupled device).

Response: The Suzhou NIR-Optics Co., Ltd., China, as referenced in our manuscript, is a high-tech company that specializes in developing near-infrared fluorescence imaging technology in the NIR-II region. The NIR-OPTICS Series III 900/1700 *In Vivo* Imaging System has been featured in a variety of articles published in the esteemed journals of *Nat. Commun.*, *Angew. Chem.* and *Adv. Mater.*, etc (examples include: *Nat. Commun.* **14**, 2950 (2023); *Angew. Chem. Int. Ed.* **62**, e202214875 (2023)). Additional details can be found on the company's website: <http://www.nir->

optics.com/m/productDe_1.html. After consulting the company's sales, we have confirmed the detector for this NIR-II imaging systems to be an InGaAs camera. We sincerely apologize for the error in the description and have thus changed the previous mention of "CCD (charge coupled device)" to "InGaAs camera".

(7) Throughout the manuscript, the authors repeatedly claimed the potential of their Y6CT-NPs in practical clinical surgery applications. While there is certain potential, it is far far away. Therefore, the authors may consider to lower the tone a bit.

Response: Thank you very much for your professional suggestions. As recommended, we have deleted some related expressions to lower the potential of Y6CT-NPs in practical clinical surgery applications.

(8) In Supplementary Information (SI), page S2, paragraph 2, the writing is somewhat random and lacks of logic. The bottom part on page S9 was not correctly displayed.

Response: The abovementioned part has been rewritten in the revised Supplementary Information. Please see the words on Page S2. Additionally, the bottom part on Page S9 was not correctly displayed probably due to the system error when the document was transformed into PDFs in the submission system. We apologize for our carelessness and will pay more attention to confirm all data are clearly and correctly shown during the manuscript submission process.

On Page S2 in the revised Supporting Information:

Structural characterization. The organic compounds synthesized were characterized for their structures and purities using ^1H and ^{13}C NMR spectroscopy with Bruker ARX 600 and ARX 500 spectrometers, using chloroform-*d* as the solvent and tetramethylsilane (TMS) as a reference. High-resolution mass spectra (HRMS) were obtained using a GCT Premier CAB 048 mass spectrometer to determine exact molecular weights. Molecular configurations were elucidated through single crystal X-ray diffractometer using Bruker smart Apex 2 and Bruker D8 Venture. **Nanoparticle characterization:** Particle size and morphology were examined using a Hitachi HT

7800 transmission electron microscope, with size distribution analyzed *via* dynamic light scattering (DLS) using an Omni NanoBrook device. **Photophysical characterization:** The UV-*vis*-NIR and photoluminescence (PL) spectra were recorded using a Shimadzu UV-2600i spectrophotometer and an FS5 Spectrofluorometer, respectively. Temperature curves were monitored using an FLIR E8-XT camera (FLIR System). **Theoretical calculations:** Weak interaction analysis from single crystal structure were performed using Multiwfn, and corresponding structure and IGM isosurfaces were generated using the VMD program. **Cellular experiment:** The absorbance of each sample was measured using a microplate reader (BioTek) for MTT assay. **NIR-II imaging:** A Series III 900/1700 In Vivo Imaging System (Suzhou NIR-Optics Co., Ltd., China) was used for image acquisition. In addition, excitation sources for photophysical tests, cellular, and *in vivo* experiments included laparoscopic-light (XD-303-80W, LED cold light source) and an 808 nm laser (CNI laser, MDL-XD-808-5W).

(9) The reviewer feels the lack of certain connection between the main text and the SI. Without intentionally reading the SI, the main text cannot be understood clearly and correctly, particularly all the experiments were given in SI. Therefore, some basic information about experiment needs to be mentioned in the main text.

Response: Thanks for your suggestions. We have included some necessary experimental information. Please see the blue words on Pages 14-16.

Reviewer #3

In this manuscript, the authors developed NIR-II nanoparticles by the nanoparticlization of Y6CT molecules and DSPE-PEG and employed such nanomaterials for imaging-guided surgery, including imaging of hepatic ischemia reperfusion and monitoring of kidney transplantation surgery. Indeed, the fluorophore used in this study has recently been published on Adv. Mater. 35, e2208229 (2023), which largely reduced the novelty to this work. In addition, the authors used white light to excite the fluorophore for imaging-guided surgery, however, the channel of naked eyes and image-guided surgery are independently used in practice. Thus, white light excitation is of less significance. Moreover, this work has some critical problems in terms of poor characterization data and lack of control experiments in imaging studies. Also, the imaging performance is far from impressive. Thus, I don't recommend its publication in this top tier journal.

Below are some suggestions that the authors might consider to improve this work.

Response: Thanks the Reviewer very much for his or her valuable suggestions. Maybe we did not express clearly. Compared to our previously reported *Adv. Mater.* 35, e2208229 (2023), the novelty of this manuscript can be highlighted as follows:

(1) The chemical structure of Y6CT in this manuscript is totally different from that of our previously reported molecules on *Adv. Mater.* 35, e2208229 (2023). FY6 in previously reported paper (*Adv. Mater.* 35, e2208229 (2023)) was solely utilized as a reference molecule. Although they share the same electron donor unit, these molecules exhibit distinctly different electron acceptors. Detailed structures have been shown below.

(2) As you said, the channel of naked eyes and image-guided surgery are indeed independently used in practice. However, the development of white-light activatable contrast agents for NIR-II bioimaging remains very important and necessary due to the following considerations: i) Compared to laser light, white-light is much cheaper and can be readily obtained from common sources, such as flashlights, LED laparoscopic light sources, and incandescent lamp, *etc.*; ii) The utilization of white-light for excitation in NIR-II bioimaging offers advantages in terms of safety, as it poses lower risks on both patients and light handlers compared to lasers, thereby reducing the potential to photo-induced injuries. In addition, employing white-light as the excitation source can prevent the photothermal effect caused by the NIR-II contrast agents during laser activation, thereby averting tissue injury and enabling prolonged monitoring of the surgical procedure. This is also confirmed by our experimental results (Supplementary Fig. 11); iii) Compared to lasers, white-light is much easier and more convenient to achieve large area which is very important during the surgery process. Moreover, the utilization of white-light can avoid the photobleaching of fluorophores and deliver a comprehensive and uniform excitation, ultimately securing highly quality imaging. Based on these aspects, we believe that the development of white-light activatable NIR-II imaging agents is of great significance.

(3) According to the Reviewer's suggestion, we incorporated indocyanine green (ICG) as a control to assess our imaging resolution. ICG, an FDA-approved dye, has been successfully utilized in NIR-II imaging-guided procedures (*Nat. Biomed. Eng.* **4**, 259-271(2019)). NIR-II vascular imaging in mice and NIR-II fluorescence imaging of the renal vasculature in white rabbits were performed and compared with Y6CT NPs.

Results demonstrated that the imaging resolution achieved with laser-activated ICG was significantly lower than that obtained with white-light-activated Y6CT-NPs in our manuscript.

1. In Supplementary Table 1, the quantum yield of Y6CT-NPS was calculated to be 16.12% based on the quantum yield of ICG in water (1%) as reference. However, most literatures reported that the quantum yield of ICG in water is ~2.5%. The authors should double-check the characterization data presented in this article.

Response: As you said, the quantum yield (QY) of ICG in water is commonly reported to be ~ 2.5%. However, the QY of ICG we presented in Supplementary Table 1 of the Supporting Information is reported be 1% in the original literature (see *Adv. Funct. Mater.* **30**, 1907093 (2019)). After careful check, we found that the 1% value was determined by the authors for wavelengths over 1000 nm utilizing IR-26 as a reference.

Additionally, in our manuscript, we used IR-26 instead of ICG as the reference to measure the QY of Y6CT-NPs. We also determined the NIR-II QY of Y6CT-NPs within the 1000-1400 nm range by employing IR26 as a reference, as illustrated in Supplementary Fig. 4 (now Supplementary Fig. 7). Our experimental results reveal a QY of 8.62% for Y6CT-NPs, indicating the superiority in NIR-II imaging. Please see the blue words on Pages 4, 5, the added Fig. S7 on Page S9, and updated Table S1 on Page S16.

On Page 4 of the revised manuscript:

“To evaluate the luminescence efficiency of Y6CT-NPs, their QY was determined using commercially available NIR-II contrast agent 4-[2-[2-chloro-3-[2-(2-phenyl-2H-thiochromen-4-yl)ethenyl]cyclohex-2-en-1-ylidene]ethylidene]-2-phenylthiochromene (IR26) (QY = 0.5%) in 1,2-dichloroethane (DCE) as the reference^{24,39}. As shown in Fig. 1e and Supplementary Fig. 7, Y6CT-NPs display an impressive QY of 16.12% in the 850-1400 nm range in aqueous solution (QY of 8.62% in the 1000-1400 nm range) using 808 nm laser as excitation source, which is ~ 32-times that of IR26 and ~ 4-times higher than our previously reported FY6-NPs (QY of

4.08% and 2.24% in the spectral ranges of 850-1400 nm and 1000-1400 nm, respectively)³⁹.”

“More importantly, Y6CT-NPs show excellent emission with a NIR-II brightness values (> 1000 nm) of 7120.1, far exceeding those of previously reported NIR-II organic small molecule nanomaterials (Fig. 1f and Supplementary Table 1).”

On Page S9 in the revised Supporting Information:

Supplementary Fig. 7. a-c The fluorescence spectra of IR26 in DCE (a), FY6-NPs (b) and Y6CT-NPs in deionized water (c) with different optical densities (ODs) under 808 nm laser excitation. d The linear fitting of the integrated PL intensity in the range of 1000-1400 nm vs. the absorbance values of Y6CT-NPs, FY6-NPs and IR26.

On Page S16 in the revised Supporting Information:

Supplementary Table 1. Photophysical characterization of previously reported organic NIR-II fluorophores³⁻²¹.

Name	Excitation light	ϵ_{\max} (L mol ⁻¹ cm ⁻¹)	λ_{em} (nm)	QY ^a (%)	Brightness (QY × ϵ)	Ref.
Y6CT-NPs	White-light	8.26 × 10⁴	947, 1030	16.12 8.62 (NIR-II)	13315.1 7120.1 (NIR-II)	This work
FY6-NPs	808 nm laser	7.64 × 10 ⁴	947, 1052	4.20	3208.8	3
BNDI-Me NPs	808 nm laser	16.4 × 10 ⁴	1104	1.40	2296.0	4
CPTIC NFs	808 nm laser	14.5 × 10 ⁴	1110	3.90	5655.0	5
BMIC-BO-4Cl NPs	880 nm laser	9.30 × 10 ⁴	1010	2.29	2129.7	6
IR-FFCHP NPs	808 nm laser	~ 1.25 × 10 ⁴	1038	7.30	912.5	7
L1013 NPs	808 nm laser	1.39 × 10 ⁴	1013	9.9	1376.1	8
TT3-oCB NPs	793 nm laser	2.07 × 10 ⁴	1062	4.60	952.2	9
TPE-BBT NPs	808 nm laser	1.40 × 10 ⁴ (DMSO)	955	31.5	4410	10
TTQiT NPs	808 nm laser	3.89 × 10 ⁴	1102	3.70	1439.3	11
TA1 NPs	808 nm laser	2.14 × 10 ⁴	893	0.08 ^b	17.12	12
HL3 dots	808 nm laser	0.93 × 10 ⁴	1125	11.70	1088.1	13
FD-1080 J-aggregate	1064 nm laser	5.00 × 10 ⁴	1370	0.54	270	14
ICG	808 nm laser	12.0 × 10 ⁴	822	1	1200	15
5H5	1064 nm laser	3.42 × 10 ⁴ (MeCN)	1069	2.6	889.2	16
CX-3	808 nm laser	5.13 × 10 ⁴ (DMSO)	1135	0.82	420.7	17
TPA-Et	808 nm laser	5.31 × 10 ⁴ (CH ₂ Cl ₂)	935	0.04 ^c	21.24	18
MB	623 nm LED	7.12 × 10 ⁴	ND	0.20	142.4	19
BAF4 NPs	1064 nm laser	~ 0.75 × 10 ⁴	ND	ND	ND	20
CCNU-1060 NPs	808 nm laser	1.60 × 10 ⁴	1065	0.3	48	21

(a) The QY was re-calculated using QY of IR26 = 0.5% in DCE as a standard. (b) ICG (NIR-II QY = 1%) in water were used as the reference for calculating the fluorescence QY of TA1 NPs. (c) Absolute QY.

2. *In both animal experiments, the authors did not provide sufficient control experiments for comparison. Such comparisons studies should involve either other fluorophores such as ICG and IR26 or other excitation methods (power laser).*

Response: According to your suggestion, we conducted NIR-II imaging of ICG in the abdominal blood vessels of mice and the renal blood vessels of rabbits. Supplementary Fig. 16 and Fig. 21 demonstrate that under white-light excitation, signals of blood vessels with ICG treatment can barely be detected. Despite the visualization of blood vessels under 808 nm laser excitation, the signal-to-noise ratio is considerably lower

compared to that of Y6CT-NPs activated by white-light. Please see the related discussion in blue words on Pages 8, 10 and 11, the added Fig. S21 on Page S15.

On Page 8 in the revised manuscript:

In particular, the NIR-II images with 1100 nm LP filter exhibit the highest resolution, characterized by its larger SNR (1.79) and narrower full width at half maximum (FWHM, 173.8 μm), surpassing those of the images with 900 or 1000 nm LP filter (Fig. 3b and Supplementary Fig. 15). Furthermore, a comparison of *in vivo* imaging was performed between Y6CT-NPs and the FDA-approved commercial dye ICG (Supplementary Fig. 16). Unfortunately, the NIR-II fluorescence signal of ICG is almost undetectable under white-light excitation. Additionally, *in vivo* imaging using ICG under 808 nm laser irradiation was also performed for comparison. Although detectable signals were captured as displayed in Supplementary Fig. 16, ICG exhibited notably lower SNR (1.12) than that achieved with white-light activated Y6CT-NPs (1.79), further affirming the superiority of Y6CT-NPs as a white-light activatable contrast agent. For the purpose of confirming the blood half-life of Y6CT-NPs, a series of blood samples were collected over time, with the fluorescence intensity of each sample subsequently measured, as depicted in Supplementary Fig. 17. The results displayed that the blood half-life of Y6CT-NPs is 1.15 h. In addition, the short-term metabolic behavior of Y6CT *in vivo* was assessed by conducting *ex vivo* biodistribution studies in mice on the 1st, 3rd, 5th, and 7th days post-administration of Y6CT-NPs (Supplementary Fig. 18). The results revealed bright NIR-II fluorescence signals in the liver and spleen, indicating the principal metabolic pathways of Y6CT-NPs through these organs. A subsequent gradual decrease of fluorescence intensity in these two organs suggested the progressive metabolism of Y6CT-NPs *via* the liver and spleen.

On Page S12 in the revised Supporting Information:

Supplementary Fig. 15. Cross-sectional intensity profiles (dark dots) and Gaussian fit (pink lines) along the pink line of abdominal vasculature in Fig. 3a with different LP filters.

On Page S12 in the revised Supporting Information:

Supplementary Fig. 16. **a** NIR-II fluorescence imaging of blood vessels of BALB/c mice in supine positions after *i.v.* injection of 100 μ L ICG under 808 nm laser and white-light irradiation. **b-d** Cross-sectional intensity profiles (dark dots) and Gaussian fit (red lines) along the pink line of abdominal vasculature with 900 nm (**b**), 1000 nm (**c**), 1100 nm (**d**) LP filters. Scale bar: 5 mm.

On Page 10-11 in the revised manuscript:

For comparison, ICG was incapable of providing detectable signals after activated by white-light (Supplementary Fig. 21). Even under 808 nm laser irradiation, ICG

exhibited poor performance compared to white-light activated Y6CT-NPs, thus once again confirming the superior imaging capabilities of Y6CT-NPs as a white-light activated contrast agent for *in vivo* imaging.

On Page S14 in the revised Supporting Information:

Supplementary Fig. 20. The signal-to-noise ratio of renal vessel (distal artery, distal vein, proximal artery, and proximal vein) of rabbits in different vascular anastomotic abnormalities corresponded to the end stage in Fig. 4.

On Page S15 in the revised Supporting Information:

Supplementary Fig. 21. a-b The diagram (a) and bright-field image (b) of kidney region of rabbit. **c-d** The fluorescence images of donor kidney vessels after intravenous injection of ICG under white-light irradiation (c) and 808 nm laser (d). Scale bar: 1 cm.

3. In Figure 3, the signal-to-noise ratio of the imaging is only 1.6. Such a low SNR is difficult to meet the requirement for imaging-guided surgery. In addition, the imaging performance in Figure 4 is poor under the white light excitation, thus really difficult to distinguish the vessels.

Response: Thank you very much for your suggestions. In Fig. 3, the SNR of the imaging by white-light activated Y6CT-NPs can be enhanced to 1.79 when using 1100 nm LP filter and the vascular structures with many tiny capillary vessels branching from larger vessels in the abdomen can be clearly mapped as depicted in Fig. 3a. As a comparison, NIR-II fluorescence imaging of blood vessels of BALB/c mice was also performed by using 808 nm laser activated ICG, which has been successfully used in clinical surgery (*Nat. Biomed. Eng.* **2019**, *4*, 259-271). Results in Supplementary Fig. 21 demonstrated an SNR value of 1.12 for ICG, indicating that white-light activated Y6CT-NPs is better than 808 nm laser activated ICG in NIR-II vascular imaging. Furthermore, the comparison between SNR values of white-light activated Y6CT-NPs and those of laser irradiated ICG were also performed for the kidney vessels of rabbits. As displayed in Supplementary Fig. 20 and Fig. 21, ICG displayed an SNR of 1.8 when used in NIR-II imaging of the donor kidney vessels upon 808 nm laser irradiation. While remarkably high SNR over 3.0 is obtained when using white-light activated Y6CT-NPs for the same NIR-II imaging. Besides, we also collected some previously reported works published on esteemed journals including *ACS Nano* **17**, 17082–17094 (2023) and *Angew. Chem. Int. Ed.* **59**, 3691-3698 (2020), in which similar SNR values are recorded to achieve NIR-II imaging. Finally, variations in signal-to-noise ratio within the same imaging setting may also arise from the utilization of different imaging processing software in the NIR-II live imaging system. We have added two new figures in the Supporting Information (Supplementary Fig. 16 and Fig. 21) and related descriptions are also added in blue words on Pages 8, 10 and 11.

On Page 8 in the revised manuscript:

In particular, the NIR-II images with 1100 nm LP filter exhibit the highest resolution, characterized by its larger SNR (1.79) and narrower full width at half maximum (FWHM, 173.8 μm), surpassing those of the images with 900 or 1000 nm LP filter (Fig. 3b and Supplementary Fig. 15). Furthermore, a comparison of *in vivo* imaging was performed between Y6CT-NPs and the FDA-approved commercial dye ICG (Supplementary Fig. 16). Unfortunately, the NIR-II fluorescence signal of ICG is almost undetectable under white-light excitation. Additionally, *in vivo* imaging using ICG under 808 nm laser irradiation was also performed for comparison. Although detectable signals were captured as displayed in Supplementary Fig. 16, ICG exhibited notably lower SNR (1.12) than that achieved with white-light activated Y6CT-NPs (1.79), further affirming the superiority of Y6CT-NPs as a white-light activatable contrast agent. For the purpose of confirming the blood half-life of Y6CT-NPs, a series of blood samples were collected over time, with the fluorescence intensity of each sample subsequently measured, as depicted in Supplementary Fig. 17. The results displayed that the blood half-life of Y6CT-NPs is 1.15 h. In addition, the short-term metabolic behavior of Y6CT *in vivo* was assessed by conducting *ex vivo* biodistribution studies in mice on the 1st, 3rd, 5th, and 7th days post-administration of Y6CT-NPs (Supplementary Fig. 18). The results revealed bright NIR-II fluorescence signals in the liver and spleen, indicating the principal metabolic pathways of Y6CT-NPs through these organs. A subsequent gradual decrease of fluorescence intensity in these two organs suggested the progressive metabolism of Y6CT-NPs *via* the liver and spleen.

On Page S12 in the revised Supporting Information:

Supplementary Fig. 15. Cross-sectional intensity profiles (dark dots) and Gaussian fit (pink lines) along the pink line of abdominal vasculature in Fig. 3a with different LP filters.

On Page S12 in the revised Supporting Information:

Supplementary Fig. 16. a NIR-II fluorescence imaging of blood vessels of BALB/c mice in supine positions after *i.v.* injection of 100 μ L ICG under 808 nm laser and white-light irradiation. b-d Cross-sectional intensity profiles (dark dots) and Gaussian fit (red lines) along the pink line of abdominal vasculature with 900 nm (b), 1000 nm (c), 1100 nm (d) LP filters. Scale bar: 5 mm.

On Page 10-11 in the revised manuscript:

For comparison, ICG was incapable of providing detectable signals after activated by white-light (Supplementary Fig. 21). Even under 808 nm laser irradiation, ICG

exhibited poor performance compared to white-light activated Y6CT-NPs, thus once again confirming the superior imaging capabilities of Y6CT-NPs as a white-light activated contrast agent for *in vivo* imaging.

On Page S14 in the revised Supporting Information:

Supplementary Fig. 20. The signal-to-noise ratio of renal vessel (distal artery, distal vein, proximal artery, and proximal vein) of rabbits in different vascular anastomotic abnormalities corresponded to the end stage in Fig. 4.

On Page S15 in the revised Supporting Information:

Supplementary Fig. 21. a-b The diagram (a) and bright-field image (b) of kidney region of rabbit. **c-d** The fluorescence images of donor kidney vessels after intravenous injection of ICG under white-light irradiation (c) and 808 nm laser (d). Scale bar: 1 cm.

4. The authors are suggested to test the luminescence efficiency as well as the quantum yield of Y6CT-NPs in other relevant solvents such as FBS and DCE.

Response: Based on your recommendation, we evaluated the quantum yield of Y6CT-NPs in fetal bovine serum (FBS). According to results in Supplementary Fig. 10, the quantum yield of Y6CT-NPs in FBS was slightly lower than that in water (14.48% vs 16.12%). Additionally, given that nanoparticles are prone to dissolve in organic solvents including 1,2-dichloroethane (DCE), test of luminescence efficiency in this solvent can not reflect the true QY value of Y6CT-NPs, and thus we preferred not to report the quantum yield of Y6CT-NPs in DCE. Please see the blue words on Page 6 and the added Fig. S10 on Page S10.

On Page 6 in the revised manuscript:

In addition, Y6CT-NPs maintained high brightness in PBS, fetal bovine serum (FBS: QY = 14.48%), and urine for 12 h at 37 °C without obvious variations, offering favorable stability for NIR-II imaging in biological systems (Supplementary Fig. 9 and Fig. 10). Furthermore, to exclude the unfavorable photothermal injury, the photothermal efficiency of Y6CT-NPs was assessed under white-light activation. As depicted in Supplementary Fig. 11, the temperature variation of Y6CT-NPs remained consistently stable without notable increase under continuous exposure for 6 min, even at high concentrations of 100 μ M, emphasizing favorable safety profile as a contrast agent for imaging purposes.

On Page S10 in the revised Supporting Information:

Supplementary Fig. 10. a-b The fluorescence spectra of Y6CT-NPs in deionized water (a) and fetal bovine serum (FBS) with different ODs under 808 nm laser excitation (b). c The linear fitting of the integrated PL intensity in the range of 850-1400 nm vs. the absorbance values of Y6CT-NPs in water and FBS.

5. The fluorophore employed in this study has recently been published on *Adv. Mater.* 35, e2208229 (2023), which largely reduced the novelty to this work.

Response: As mentioned in our response 3 to question 3, the fluorophore Y6CT, under investigation here, has not yet been employed in NIR-II imaging, and is totally different from our published molecule FY6 in *Adv. Mater.* 35, e2208229 (2023). FY6 in previously reported paper (*Adv. Mater.* 35, e2208229 (2023)) was solely utilized as a reference molecule. Their chemical structures are depicted as follow for your convenience of comparison.

Reviewers' Comments:

Reviewer #1:

Remarks to the Author:

Since the authors have revised the manuscript to address all comments raised by the reviewer, I think it could be acceptable for publication now.

Reviewer #2:

Remarks to the Author:

The authors have addressed my concerns on the original manuscript. However, hope the authors can develop efficient pure SWIR materials, because using the weak tail of a NIR-I material is not a right solution for SWIR imaging applications.

Reviewer #3:

Remarks to the Author:

In the revised manuscript, although the authors have conducted additional experiments, some critical issues are still not addressed. More importantly, there are some duplicative data appeared in this revised manuscript, which made the reviewer suspect and rethink the reliability of this manuscript. Therefore, I do not recommend this manuscript for publication in Nature Communication.

- 1) In page 8 of main text, the author claimed that Y6CT-NPs had negligible organ and hematological toxicity in mice, and provided the blood biochemical data in Supplementary Table 4. In Table S4, BUN, creatinine and UA for the control mice and mice injected with Y6CT-NPs were listed. In page 10 of main text, the author claimed that the negligible systemic toxicity of Y6CT-NPs was confirmed in rabbit via the histological investigation of organs and hematological analysis. In Table S6, the detailed ALB, BUN, creatinine and UA values are also listed. However, the values of ALB, BUN, creatinine and UA in mice and in rabbit are completely the same (ALB:38.642; BUN: 19.426/19.113; Crea:89.483/88.847; UA: 3.642/3.187), even the decimal points are also the same. That is impossible to obtain duplicative values for mice and rabbits, as there are different species. Therefore, I suspected and rethink the reliability of this manuscript.
- 2) The authors used ICG as a standard agent and did the control experiments for comparison. However, such experiments were only conducted in healthy mice, but not in model mice. Thus, those comparisons are insufficient. The authors should use the same dosage of ICG and Y6CT-NPs to imaging the vessel in both hepatic ischemia reperfusion and kidney transplantation surgery.
- 3) In this revised manuscript, the detailed explanation for the high quantum yield of Y6CT is not provided. The authors only explained its high brightness. However, detailed mechanisms from photophysics behind this are missing.
- 4) A lot of details are missing in terms of the agent's concentration and power density used. For example, the authors used ICG as a control agent for imaging blood vessels in mice. But the detailed concentrations are not provided. Moreover, power density are missing in lots of figure captions.

Point-by-point responses to the Reviewers' comments:

Reviewer #1

General comments: Since the authors have revised the manuscript to address all comments raised by the reviewer, I think it could be acceptable for publication now.

General answers: We sincerely thank the Reviewer very much for his or her time to review our manuscript, and finally agreeing to accept our work.

Reviewer #2

General comments: The authors have addressed my concerns on the original manuscript. However, hope the authors can develop efficient pure SWIR materials, because using the weak tail of a NIR-I material is not a right solution for SWIR imaging applications.

General answers: Thanks the Reviewer very much for his or her positive comments. Yes, the development of efficient pure SWIR materials is crucial for enhancing in vivo SWIR imaging applications. We will develop such materials in our forthcoming research. Furthermore, in this work, Y6CT-NPs exhibit dual emission peaks at 947 and 1030 nm with a NIR-II brightness values (> 1000 nm) of 7120.1, which can meet the imaging requirements of NIR-II region to a certain extent.

Reviewer #3

General comments: In the revised manuscript, although the authors have conducted additional experiments, some critical issues are still not addressed. More importantly, there are some duplicative data appeared in this revised manuscript, which made the reviewer suspect and rethink the reliability of this manuscript. Therefore, I do not recommend this manuscript for publication in Nature Communication.

General answers: Thanks the Reviewer very much for his or her some positive comments and some critical issues.

Q1: In page 8 of main text, the author claimed that Y6CT-NPs had negligible organ and hematological toxicity in mice, and provided the blood biochemical data in Supplementary Table 4. In Table S4, BUN, creatinine and UA for the control mice and mice injected with Y6CT-NPs were listed. In page 10 of main text, the author claimed that the negligible systemic toxicity of Y6CT-NPs was confirmed in rabbit via the histological investigation of organs and hematological analysis. In Table S6, the detailed ALB, BUN, creatinine and UA values are also listed. However, the values of ALB, BUN, creatinine and UA in mice and in rabbit are completely the same (ALB:38.642; BUN:19.426/19.113; Crea:89.483/88.847; UA: 3.642/3.187), even the decimal points are also the same. That is impossible to obtain duplicative values for mice and rabbits, as there are different species. Therefore, I suspected and rethink the reliability of this manuscript.

A1: Thanks the Reviewer very much for his or her careful reviewing our manuscript. We sincerely apologize for the data error in the description of the Supplementary Table 4 due to our incaution. We have re-corrected and rechecked to ensure the accuracy of the data. In addition, we have attached the raw data as the evidence in the attachment.

Q2: The authors used ICG as a standard agent and did the control experiments for comparison. However, such experiments were only conducted in healthy mice, but not in model mice. Thus,

those comparisons are insufficient. The authors should use the same dosage of ICG and Y6CT-NPs to imaging the vessel in both hepatic ischemia reperfusion and kidney transplantation surgery.

A2: Thanks the Reviewer very much for his or her valuable suggestions. As you said, we have conducted control experiments with ICG as a standard agent in healthy mice and rabbit blood vessels. These control experiments showed that ICG cannot display any fluorescence signal under white-light irradiation. Therefore, redundant control experiments are nonessential in hepatic ischemia reperfusion and kidney transplantation surgery, which does not comply with the 3Rs principle (replacement, reduction and refinement) of animal ethics.

Q3: In this revised manuscript, the detailed explanation for the high quantum yield of Y6CT is not provided. The authors only explained its high brightness. However, detailed mechanisms from photophysics behind this are missing.

A3: In the crystal analysis and theoretical calculations of the manuscript, we have offered a detailed explanation for the high quantum yield of Y6CT-NPs. Y6CT-NPs demonstrates a *J*-aggregation stacking mode and strong intermolecular D-A interactions, facilitating constructive coupling of excited-state transition dipoles. Therefore, Y6CT-NPs exhibits high fluorescent quantum yield. As recommended, we have been also added more detailed discussion in blue words on Page 7.

Q4: A lot of details are missing in terms of the agent's concentration and power density used. For example, the authors used ICG as a control agent for imaging blood vessels in mice. But the detailed concentrations are not provided. Moreover, power density are missing in lots of figure captions.

A4: Thanks the Reviewer very much for his or suggestion. In accordance with your suggestions, we have included the missing details about the concentration of the agent and the power density used in the figure captions. Many of our experimental details are shown in the experimental methods, such as the agent's concentration and power density of excitation light.

Reviewers' Comments:

Reviewer #3:

Remarks to the Author:

The authors have addressed all the issues, thus it can be accepted now.